# Competition for synaptic building blocks shapes synaptic plasticity

Jochen Triesch[1,2]*, Anh Duong Vo[1,2], Anne-Sophie Hafner[3]

[1]Frankfurt Institute for Advanced Studies, Frankfurt am Main, Germany; [2]Goethe University, Frankfurt am Main, Germany; [3]Max-Planck Institute for Brain Research, Frankfurt am Main, Germany

**Abstract** Changes in the efficacies of synapses are thought to be the neurobiological basis of learning and memory. The efficacy of a synapse depends on its current number of neurotransmitter receptors. Recent experiments have shown that these receptors are highly dynamic, moving back and forth between synapses on time scales of seconds and minutes. This suggests spontaneous fluctuations in synaptic efficacies and a competition of nearby synapses for available receptors. Here we propose a mathematical model of this competition of synapses for neurotransmitter receptors from a local dendritic pool. Using minimal assumptions, the model produces a fast multiplicative scaling behavior of synapses. Furthermore, the model explains a transient form of heterosynaptic plasticity and predicts that its amount is inversely related to the size of the local receptor pool. Overall, our model reveals logistical tradeoffs during the induction of synaptic plasticity due to the rapid exchange of neurotransmitter receptors between synapses.
DOI: https://doi.org/10.7554/eLife.37836.001

## Introduction

Simple mathematical models of Hebbian learning exhibit an unconstrained growth of synaptic efficacies. To avoid runaway dynamics, some mechanism for limiting weight growth needs to be present. There is a long tradition of addressing this problem in neural network models using synaptic normalization rules (*von der Malsburg, 1973*; *Oja, 1982*; *Miller and MacKay, 1994*; *Wu and Yamaguchi, 2006*; *Lazar et al., 2009*). Obviously, in order to keep up with the pace of synaptic changes due to Hebbian plasticity, normalization mechanisms must act sufficiently fast. Slow homeostatic synaptic scaling mechanisms (*Turrigiano et al., 1998*) may therefore be ill-suited for ensuring stability (*Wu and Yamaguchi, 2006*; *Zenke et al., 2013*; *Chistiakova et al., 2015*). A particularly interesting fast normalization rule scales synapses multiplicatively such that the sum of synaptic weights remains constant. Attractive features of such a rule, next to its conceptual simplicity, are that the relative strength of synapses is maintained and that in combination with Hebbian mechanisms it naturally gives rise to lognormal-like weight distributions as observed experimentally (*Song et al., 2005*; *Loewenstein et al., 2011*; *Zheng et al., 2013*; *Miner and Triesch, 2016*). While such normalization mechanisms are not considered biologically implausible, their link to neurobiological experiments has been tenuous.

In a recent review, *Chistiakova et al. (2015)* argue that so-called heterosynaptic plasticity (*Lynch et al., 1977*; *Bailey et al., 2000*; *Jedlicka et al., 2015*; *Antunes and Simoes-de-Souza, 2018*) may be a prime candidate for such a fast synaptic normalization scheme. The term 'heterosynaptic' plasticity is used in contrast to the much more widely studied 'homosynaptic' plasticity, where changes occur in a stimulated synaptic pathway. In contrast, heterosynaptic plasticity refers to changes in synaptic efficacies that occur in an unstimulated pathway after the stimulation of a neighboring pathway. The most common form of heterosynaptic plasticity has a homeostatic nature: if synapses in stimulated pathways potentiate, then this is accompanied by a depression of

*For correspondence:
triesch@fias.uni-frankfurt.de

**Competing interests:** The authors declare that no competing interests exist.

unstimulated pathways. Conversely, if synapses in stimulated pathways depress, this is accompanied by a potentiation of unstimulated pathways. A classic example of this has been observed in intercalated neurons of the amygdala (*Royer and Paré, 2003*).

Interestingly, such homeostatic regulation is also consistent with findings at the ultra-structural level. The physical size of a synapse, in particular the surface area of the postsynaptic density (PSD), is commonly used as a proxy for a synapse's efficacy (*Chen et al., 2015*; *Bartol et al., 2015*). *Bourne and Harris (2011)* have observed coordinated changes in PSD surface areas of dendritic spines in the hippocampus after LTP induction. They report that increases in the PSD surface areas of some synapses or the creation of new synapses are balanced by decreases of PSD surface areas of other synapses or their complete elimination such that the total amount of PSD surface area stays approximately constant. Recent findings support the idea that such regulation may occur at the level of individual dendritic branches (*Barnes et al., 2017*).

A proxy of synaptic efficacy that is more precise than PSD surface area is the number of AMPA receptors (AMPARs) inside the PSD. AMPARs are glutamate-gated ion channels responsible for most fast excitatory transmission in the vertebrate brain. During various forms of plasticity the number of these receptors at synapses is modified, leading to changes in synaptic efficacies, reviewed by *Chater and Goda (2014)*. Therefore, a full understanding of synaptic plasticity requires a careful description of the mechanisms that regulate AMPAR numbers in synapses.

Here we show how the behavior of keeping the sum of synaptic efficacies approximately constant on short time scales naturally arises from a generic model in which individual synapses compete for a limited supply of synaptic building blocks such as AMPARs or other protein complexes that are necessary to stabilize AMPARs inside the PSD. We assume that there is a local dendritic store of these building blocks and that they enter and leave dendritic spines in a stochastic fashion. The model predicts that the redistribution of synaptic efficacies should act multiplicatively, as is often assumed in *ad hoc* normalization models. We also show that this model naturally gives rise to a homeostatic form of heterosynaptic plasticity, where synapses grow at the expense of other synapses. To this end, we introduce a model of homosynaptic LTP describing the time course of the incorporation of new receptors and slots during LTP induction. Finally, we quantify the scale of spontaneous synaptic efficacy fluctuations due to the fast stochastic exchange of AMPARs between the dendritic pool and postsynaptic receptor slots. We show that small synapses exhibit relatively stronger efficacy fluctuations, which are further accentuated if the local receptor pool is small. Overall, the model reveals how the dynamic behavior of neurotransmitter receptors plays an important role in shaping synaptic plasticity.

## Results

### Formulation of the model

The architecture of the model is shown in *Figure 1*. We consider a piece of dendrite with $N \in \mathbb{N}$ synaptic inputs. Each synapse is characterized by two variables. First, each synapse $i \in 1, \ldots, N$ has a number of slots $s_i \in \mathbb{R}^{\geq 0}$ for neurotransmitter receptors. Second, at any time a certain number of slots $w_i \in \mathbb{R}^{\geq 0}$ actually contain a receptor. $w_i$ determines the current weight or efficacy of a synapse. We assume that the PSD cannot hold more functional receptors than there are slots, that is, $w_i \leq s_i$. At biological synapses AMPARs are clustered inside PSDs into nanodomains of about 70 nm that contain on average 20 receptors (*Nair et al., 2013*). Interestingly, those postsynaptic nanodomains are aligned with presynaptic release sites forming so-called nanocolunms. It is noteworthy that AMPARs have low affinity for glutamate such that receptors outside of nanodomains are unlikely to participate in synaptic transmission (*Liu et al., 1999*; *Biederer et al., 2017*). Next to receptors in the synapses, the neuron maintains a pool of receptors freely diffusing at the neuron surface and ready to be stabilized inside nanodomains. The size of this pool is denoted $p \in \mathbb{R}^{\geq 0}$. Note that for mathematical convenience we here consider the $s_i$, $w_i$ and $p$ to be real numbers that can take non-integer values. In the stochastic version of the model introduced below these will be natural numbers.

Receptors can transition from the pool to empty slots in a synapse or detach from such a slot and return into the pool with rates $\alpha \in \mathbb{R}^{>0}$ and $\beta \in \mathbb{R}^{>0}$, respectively. Receptors in the pool are removed with a rate $\delta \in \mathbb{R}^{>0}$ corresponding to internalization of the receptors from the cell surface

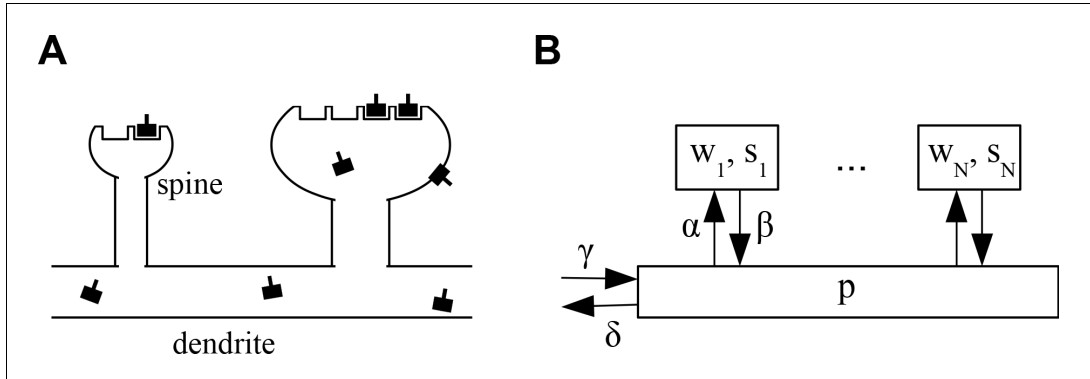

**Figure 1.** Architecture of the model. (**A**) Sketch of the architecture of the model. Neurotransmitter receptors, e.g. AMPA receptors, are trafficked through the dendrite and bind to 'slots' inside of dendritic spines. The efficacy of a synapse is assumed to be proportional to the number of receptors attached to its slots. (**B**) Abstract description of the stochastic process indicating the rates at which receptors move in and out of slots in the synapses and the receptor pool in the dendrite. See text for details.
DOI: https://doi.org/10.7554/eLife.37836.002

(endocytosis). To counteract this loss, new receptors are added at a rate $\gamma \in \mathbb{R}^{>0}$ and injected into the pool corresponding to externalization of the receptors to the cell surface (exocytosis). In the limit of large receptor numbers, the dynamics of the system can be described by the following system of coupled ordinary nonlinear differential equations:

$$\dot{w}_i = -\beta w_i + \alpha p(s_i - w_i), \; i = 1, \ldots, N \tag{1}$$

$$\dot{p} = -\delta p + \gamma + \sum_i \beta w_i - \sum_i \alpha p(s_i - w_i). \tag{2}$$

In the first equation, $-\beta w_i$ describes the return of receptors from synapse $i$ into the pool. The term $\alpha p(s_i - w_i)$ describes the binding of receptors from the pool to empty slots in synapse $i$, which is assumed to be proportional to both the number of receptors in the pool and the number of free slots in the synapse. In the second equation, $-\delta p$ describes the deletion of receptors from the pool, $\gamma$ represents the gain of new receptors, $\sum_i \beta w_i$ describes the return of receptors from the synapses into the pool, and finally $-\sum_i \alpha p(s_i - w_i)$ describes the loss of receptors from the pool which bind to free slots in the synapses. Together, this is a system of $N + 1$ coupled ordinary nonlinear differential equations. It is nonlinear, because the equations contain product terms of the state variables, in particular the $pw_i$ terms.

The model can be interpreted in different ways. Its generic interpretation is that the 'receptors' of the model are AMPA receptor (AMPAR) complexes composed of AMPARs and transmembrane AMPAR regulatory proteins (TARPs) such as stargazin. The 'slots' are postsynaptic density structures comprising membrane-associated guanylate kinase (MAGUK) proteins such as PSD-95 attached to the postsynaptic membrane, which stabilize AMPARs in the postsynaptic density (PSD) (*Hafner et al., 2015*; *Schnell et al., 2002*; *Sumioka et al., 2010*). Inside the synapses PSD-95 proteins are highly packed (roughly 300 molecules per PSD) (*Kim and Sheng, 2004*) and largely immobile (*Sturgill et al., 2009*). When a receptor enters a synapse binding to one or more immobile PSD-95 proteins results in receptor immobilization. In this generic interpretation of the model, the pool of receptors is the set of AMPARs that diffuse in the plasma membrane and that are captured by the slots. Addition of receptors to the pool then subsumes (some or all of) the processes that assemble AMPARs and prepare them for the insertion into slots: assembly of the receptors from the component subunits, trafficking, attachment of TARPs, externalization, and potentially phosphorylation. Removal from the pool similarly subsumes the set of reverse processes. Several variations of this generic interpretation are possible depending on what exactly we would like to associate with the 'receptors' in the model: AMPARs, AMPAR+TARP complexes, AMPAR+TARP complexes that have

already been exocytosed, phosphorylated, etc. Essentially, our model is a two-step model (production and insertion), but we leave it open for interpretation, what steps in the full chain of events are considered the 'production' (subsumed in rate $\gamma$) and which steps are considered the 'insertion' (subsumed in rate $\alpha$).

Evidently, receptor slots themselves must also be stabilized inside the PSD somehow. A second, maybe somewhat counter-intuitive, interpretation of the model is therefore that it describes the binding and unbinding of receptor slots to what one might consider a *slot for a receptor slot* or simply *slot-for-a-slot*. In this interpretation of the model, the 'receptors' in the description above are actually the PSD-95 slot proteins and the 'slots' are slots-for-a-slot to which the PSD-95 proteins can attach. The model then describes the trafficking of PSD-95 into and out of the PSD, assuming that available AMPAR complexes are quickly redistributed among PSD-95 slots (compared to the time scale of addition and removal of these PSD-95 slots to the PSD). This interpretation may be particularly useful if the supply of PSD-95 is the limiting factor in determining the number of functional AMPARs bound inside the PSD (*Schnell et al., 2002*). We leave open the question what exactly the slots-for-a-slot might be. It is clear however, that PSD-95 molecules can form stable lattices inside the PSD such that PSD-95 proteins could act as slots for other PSD-95 proteins.

Interestingly, the analysis of the model presented in the following does not depend on which interpretation is chosen. The only additional assumption we will make is a separation of time scales between the fast trafficking of the 'receptors' into and out of the 'slots' and the slow addition and removal of receptors to the pool. Our main results only depend on this qualitative feature of the model. For the first generic interpretation of the model the assumption of a separation of time scales appears justified. If we interpret the receptor pool of the model to comprise AMPARs that have been exocytosed and diffuse in the cell membrane, then the half-life of an AMPAR in the pool is of the order of 10 min suggesting $\delta^{-1} = 10 \, \text{min} / \ln 2 \approx 14 \, \text{min}$ (*Henley and Wilkinson, 2013*; *Henley and Wilkinson, 2016*). In contrast, the time an AMPAR stays inside the PSD, which we interpret as the time the AMPAR is bound to a slot, appears to be of the order of maybe 30 s (*Ehlers et al., 2007*), suggesting $\beta^{-1} = 30 \, \text{s} / \ln 2 \approx 43 \, \text{s}$. We summarize these and other parameters of the model in *Table 1*. Regarding the second, slots-for-a-slot, interpretation of the model, we note that the half-life of PSD-95 residing inside the synapse is of the order of 5 h (*Sturgill et al., 2009*), implying $\beta^{-1} \approx 5 \, \text{h} / \ln 2 \approx 7 \, \text{h}$. In contrast, the global half-life of PSD-95 has been estimated to be 3.67 days (*Cohen et al., 2013*), implying $\delta^{-1} = 3.67 \, \text{d} / \ln 2 \approx 5.30 \, \text{d}$. In either case, the assumption of a separation of time scales appears justified.

## Competition for synaptic building blocks induces multiplicative scaling

We begin our analysis by finding the stationary solution of the system of coupled differential equations defined by *Equations 1 and 2*. First, it is convenient to introduce the total number of synaptic slots $S \equiv \sum_i s_i$ and the total number of docked receptors or total synaptic weight $W \equiv \sum_i w_i$ and note that its time derivative is $\dot{W} = \sum_i \dot{w}_i$. This allows us to rewrite *Equation 2* as:

$$\dot{p} = -\delta p + \gamma + \beta W - \alpha p (S - W) \, . \tag{3}$$

To find the fixed point solution $p^\infty, w_i^\infty$ with $W^\infty = \sum_i w_i^\infty$, we set the time derivatives to zero,

**Table 1.** Standard parameters of the model.

|          | Value              | Description          | Reference                                              |
| -------- | ------------------ | -------------------- | ------------------------------------------------------ |
| $\beta$  | $(43 \, \text{s})^{-1}$ | unbinding rate from slots | *Henley and Wilkinson (2013)*; *Henley and Wilkinson (2016)* |
| $\delta$ | $(14 \, \text{min})^{-1}$ | internalization rate | *Ehlers et al. (2007)*                                 |
| $\phi$   | 2.67               | relative pool size   | M. Renner, personal communication                      |
| $F$      | unknown            | filling fraction     | set by hand to {0.5, 0.7, 0.9}                         |
| $\gamma$ | unknown            | externalization rate | set via *Equation 12* to achieve desired $\phi$        |
| $\alpha$ | unknown            | binding rate to slots | set via *Equation 13* to achieve desired $F$           |

DOI: https://doi.org/10.7554/eLife.37836.003

that is, we require $\dot{w}_i = 0 \; \forall i$ and $\dot{p} = 0$ above. Inserting the first condition into **Equation 1** and summing over $i$ yields:

$$0 = -\beta W^\infty + \alpha p^\infty (S - W^\infty).$$

(4)

Similarly, setting $\dot{p} = 0$ in **Equation 3** gives:

$$0 = -\delta p^\infty + \gamma + \beta W^\infty - \alpha p^\infty (S - W^\infty).$$

(5)

Adding **Equation 4** to **Equation 5** then gives the solution for $p^\infty$:

$$p^\infty = \frac{\gamma}{\delta}.$$

(6)

The simple and intuitive result is therefore that the total number of receptors in the pool in the steady state is given by the ratio of the externalization rate $\gamma$ and the internalization rate $\delta$. Specifically, the presence of many receptors in the pool requires $\gamma \gg \delta$.

We now solve for the steady state solutions $w_i^\infty$ of the $w_i$ by again setting $\dot{w}_i = 0$ in **Equation 1** and using **Equation 6** to give:

$$w_i^\infty = \frac{1}{1 + \frac{\beta\delta}{\alpha\gamma}} s_i \equiv F s_i.$$

(7)

Importantly, we find $w_i^\infty \propto s_i$, that is, in the steady state the weights of synapses are proportional to the numbers of slots they have. The constant of proportionality is a *filling fraction* and we denote it by $F$. Interestingly, the filling fraction $F$ is independent of the number of receptor slots. **Figure 2A** plots $F$ as a function of the ratio of the four rate constants $(\beta\delta)/(\alpha\gamma)$. We refer to this quantity as the removal ratio, because it indicates the rates of the processes that remove receptors from the slots relative to the rates of the processes that add them to slots. Note that a filling fraction close to one requires $\beta\delta \ll \alpha\gamma$.

Summing **Equation 7** over $i$ reveals that $W^\infty = FS$, so we can also write:

$$w_i^\infty = \frac{s_i}{S} W^\infty,$$

(8)

where $s_i/S$ is the relative contribution of synapse $i$ to the total number of slots. Note that if the filling

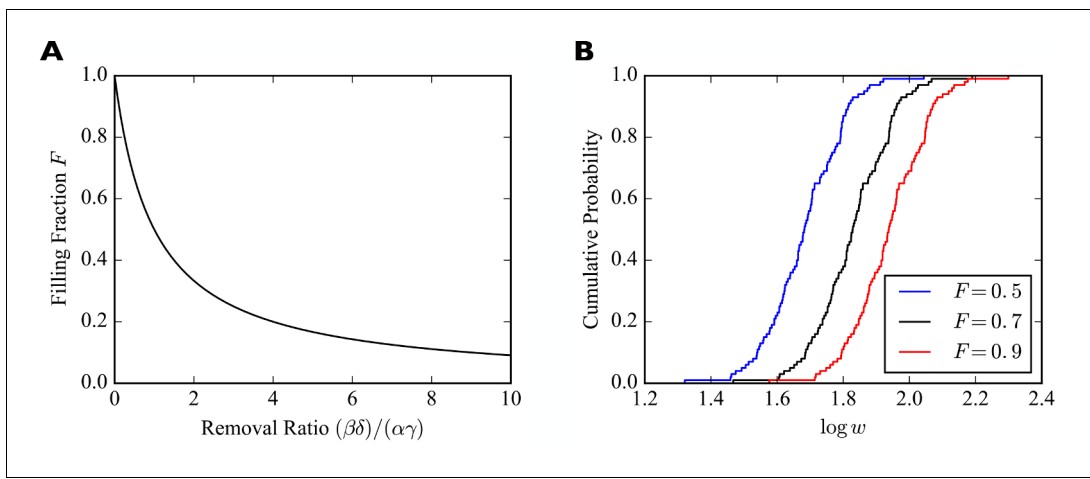

**Figure 2.** Synaptic filling fraction. (**A**) Filling fraction $F$ as a function of the removal ratio $(\beta\delta)/(\alpha\gamma)$. (**B**) Example empirical cumulative distribution functions (CDFs) of the numbers of receptors bound in individual synapses for fixed numbers of slots drawn from a lognormal distribution and three different filling fractions $F$. The simulated piece of dendrite has 100 synapses and 100 receptor slots per synapse on average.
DOI: https://doi.org/10.7554/eLife.37836.004

fraction changes, say, due to an increase in receptor externalization or a change in any of the other parameters, the *relative* strength of two synapses in the steady state is unaffected:

$$\frac{w_i^\infty}{w_j^\infty} = \frac{s_i}{s_j} = \text{const.} \tag{9}$$

Therefore, all synaptic efficacies will be scaled multiplicatively by the same factor.

Thus, the analysis so far reveals a first prediction of the model (compare *Table 2*, *Filling Fraction*): Under basal conditions synapses in the local group have identical filling fractions. A first corollary from this prediction is that the ratio of two synapses' efficacies in the steady state is given by the ratio of their numbers of receptor slots. A second corollary from this prediction is that when one (or more) of the transition rates changes, all synaptic efficacies are scaled multiplicatively.

To illustrate the effect of multiplicative scaling of synaptic efficacies, we consider a piece of dendrite with $N = 100$ afferent synapses. The number of receptor slots $s_i$ in these synapses are drawn from a lognormal distribution with mean 1.0 and standard deviation 0.2 and subsequently scaled such that there are 100 slots per synapse on average. We consider three different filling fractions $F \in \{0.5, 0.7, 0.9\}$. The empirical cumulative distribution functions (CDFs) of the common (decadic) logarithms of the $w_i$ are shown in *Figure 2B*. The horizontal shifting of the empirical CDFs illustrates the multiplicative scaling of the individual synaptic efficacies.

The total number of receptors in the system in the steady state $R^\infty$ is given by the sum of the number of receptors in the pool and the number of receptors attached to slots. Combining the above results, we find:

$$R^\infty = p^\infty + W^\infty = p^\infty + FS = \frac{\gamma}{\delta} + \frac{1}{1 + \frac{\beta\delta}{\alpha\gamma}}S. \tag{10}$$

In particular, the total number of receptors in the steady state depends on the total number of slots.

In the case of AMPARs, the total number of surface receptors, receptor density, or the number of slots per synapse still remain unknown. Moreover, it is likely that those numbers will vary depending on neuron type and developmental state. However, single particle tracking experiments from the laboratory of Antoine Triller performed on mature hippocampal cultured neurons provide valuable insights into the proportion of exocytosed receptors immobilized in dendritic spines in this particular system. Specifically, recent data suggest that 28% of surface AMPARs are immobilized at synapses while the remaining 72% reside in the pool of extrasynaptic receptors (Marianne Renner, personal communication). Since mature hippocampal cultured neurons are known to exhibit homeostatic and long-term plasticity, we decided to use those numbers for our simulations. Thus, we define the relative pool size $\phi$ as:

$$\phi = \frac{p^\infty}{W^\infty} = \frac{0.72R^\infty}{0.28R^\infty} \approx 2.67. \tag{11}$$

**Table 2.** Summary of model predictions.
Further predictions are mentioned in the Discussion.

| Prediction | Explanation |
| --- | --- |
| Filling fraction | Synapses in a local group have identical filling fractions in the basal state. |
| Pool Size | Manipulation of local pool size scales synapses multiplicatively. |
| Sensitivity | Filling fraction is most sensitive when pool size matches slot numbers. |
| Heterosynaptic I | High pool size and filling fraction reduce heterosynaptic plasticity. |
| Heterosynaptic II | Heterosynaptic plasticity is only transient. |
| Homosynaptic | Pool size and filling fraction modulate homosynaptic plasticity. |
| Fluctuations | Spontaneous efficacy fluctuations are bigger for small synapses. |

DOI: https://doi.org/10.7554/eLife.37836.005

The relative pool size $\phi$ together with the filling fraction $F$ determine the unknown externalization rate $\gamma$ and the rate of binding to receptor slots $\alpha$. Specifically, using $W^\infty = FS$ and $p^\infty = \gamma/\delta$, we find:

$$\gamma = \delta p^\infty = \delta FS\phi. \tag{12}$$

Furthermore, by combining this with the implicit definition of $F$ from *Equation 7* we can solve for $\alpha$ to obtain:

$$\alpha = \frac{\beta\delta}{\gamma}\frac{F}{1-F} = \frac{\beta}{\phi S(1-F)}. \tag{13}$$

We can identify the term $S(1-F)$ as the total number of empty receptor slots in the system. The intuitive interpretation of the result is therefore that the binding rate $\alpha$ will be big compared to the unbinding rate $\beta$ if the number of empty slots and the relative pool size are small. Using the definition of $\phi$ we can also rewrite the expression for the total number of receptors as $R^\infty = (1+\phi)FS$.

The above results fully describe the system after it had sufficient time to reach its equilibrium. On a shorter time scale, however, the system may transiently assume different quasi-stationary states, because receptor addition and removal are slow compared to receptor binding and unbinding to and from slots. In the following, we consider the short-term behavior of the model on time scales where the total number of receptors is approximately constant. This will allow us to reveal, among other things, a transient form of heterosynaptic plasticity.

## Fast redistribution of receptors between synapses is multiplicative

To study the redistribution of receptors on a fast time scale, we exploit the fact that the processes of receptor externalization and internalization are slow compared to the attaching and detaching of receptors to and from slots. For instance, the time that an AMPAR remains in the cell membrane is of the order of ten minutes while the time it resides inside the PSD is of the order of half a minute. A reasonable approximation on short times scales is therefore to neglect the production and removal terms in *Equation 2*. In this case, the total number of receptors $R \equiv W + p$ is constant, as can be seen by removing the $-\delta p$ and $+\gamma$ terms from *Equation 2*, and adding *Equation 1*, summed over all $i$, which gives $\dot{p} + \dot{W} = \dot{R} = 0$. In the Methods we show that the steady state solution on the fast time scale is then given by:

$$W^* = \frac{1}{2}(S+R+\rho) - \sqrt{\frac{1}{4}(S+R+\rho)^2 - RS}, \tag{14}$$

where we have introduced $\rho \equiv \beta/\alpha$ as a short hand for the ratio of the rates through which receptors leave and enter the synaptic slots. We define the corresponding short-term steady-state filling fraction as $F^* = W^*/S$. Importantly, the short-term filling fraction $F^*$ is identical for all synapses. $F^*$ can also be expressed as a function of the steady state pool size $p^* = R - W^*$ on the fast time scale, leading to a simple expression for the steady state efficacy $w_i^*$ of synapse $i$ on the fast time scale:

$$w_i^* = F^* s_i = \frac{p^*}{p^* + \rho}s_i. \tag{15}$$

In the full model, this solution is assumed only transiently, because receptors can still enter and leave the system. If the number of receptors were held constant ($\gamma = 0$ and $\delta = 0$), then $F^*$, $p^*$, and the $w_i^*$ would describe the solution on long time scales.

The finding that the short-term steady-state filling fraction is identical for all synapses is analogous to the solution for the long term filling fraction $F$ derived in *Equation 7*, which is also the same for all synapses. This implies a second prediction of the model (compare *Table 2*, *Pool Size*): When the size of the local receptor pool is manipulated, all synaptic efficacies are scaled multiplicatively.

In *Figure 3* we show the behavior of $F^*$ as a function of $\rho$ for different combinations of total number of slots $S$ and total number of receptors $R$. For high values of $\rho$ the filling fraction $F^*$ always goes to zero. For $\rho$ approaching zero, $F^*$ achieves a maximum value which depends on whether there are fewer or more receptors than slots in the system. If there are more receptors than slots then $F^*$ approaches one. If there are fewer receptors than slots then $F^*$ approaches the ratio of receptors to

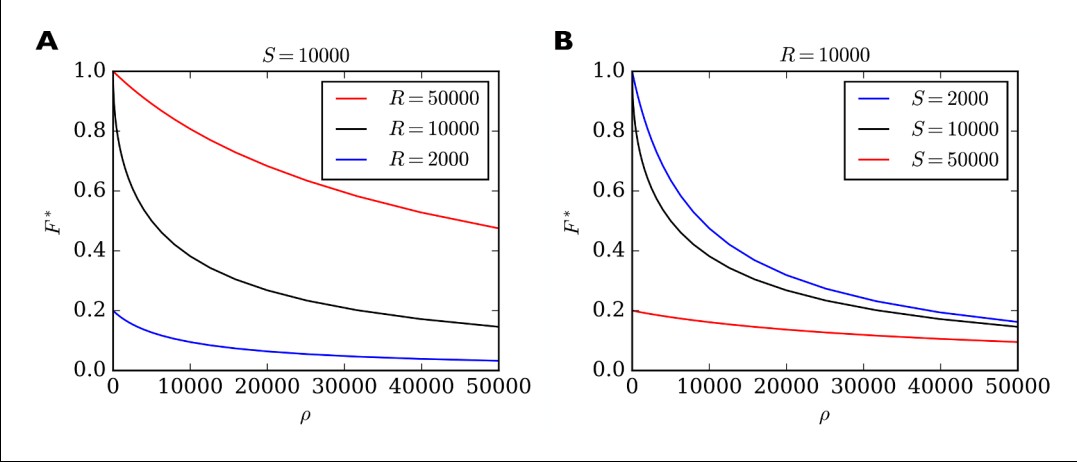

**Figure 3.** Filling fraction $F^*$ in the short-term approximation of constant receptor number as a function of the ratio of transition rates $\rho = \beta/\alpha$ for different combinations of $R$ and $S$. (A) $F^*$ for a fixed number of $S = 10\,000$ slots and three different receptor numbers as a function of $\rho$. (B) For fixed number of $R = 10\,000$ receptors and three different numbers of slots. Note that $F^*$ reacts particularly sensitively to changes in $\rho$ when $\rho$ is small and when $R = S$ (black curves). In this regime, small changes to, say, the rate of detaching from slots $\beta$ have a great influence on the filling fraction. In all cases, the shown solution $F^*$ is only transient. Eventually the filling fraction will assume its steady state value $F$ given by **Equation 7**.

DOI: https://doi.org/10.7554/eLife.37836.006

slots in the system. In general, we find that the maximum short-term filling fraction for $\rho \to 0$ is given by $F^*_{\max} = min\{1, R/S\}$. In particular, a high filling fraction can only be achieved if $R > S$, that is, there must be more receptors than slots in the system. On the other hand, $F^*$ is most sensitive to changes in $\rho$ when $R = S$. This can be seen by the steep negative slope of the black curves in **Figure 3** for small values of $\rho$. In fact, for $R = S$ the derivative diverges, that is, $F^*$ reacts extremely sensitively to changes in $\rho$ (see Methods for details). We therefore note another prediction (compare **Table 2**, *Sensitivity*): On short time scales the filling fraction reacts most sensitively to changes in binding/unbinding rates if the total number of receptors matches the total number of receptor slots.

To illustrate the fast redistribution of receptors, we consider a sudden change in the pool size. In our generic interpretation of the model, this corresponds to the sudden externalization or internalization of AMPARs. To study the effect of such a manipulation, we discretize the full dynamic equations using the Euler method and solve them numerically. For illustration, we consider a piece of dendrite with just three synapses with 40, 60, and 80 slots, whose pool size is changed abruptly (**Figure 4**). Parameters are set to achieve a filling fraction of $F = 0.9$ and a relative pool size $\phi = 2.67$. After 2 min, the number of receptors in the pool is either doubled (solid lines) or set to zero (dotted lines). In response, all synapses are rapidly scaled up or down multiplicatively. The new equilibrium is only transient, however. On a slower time scale the system returns to its starting point as the slow externalization and internalization processes drive the system back to its steady state solution $w_i^\infty, p^\infty$.

The fast equilibration process to a transient steady state also naturally gives rise to a homeostatic form of heterosynaptic plasticity. When, e.g., the number of receptor slots in some synapses is quickly increased, then receptors are redistributed such that the efficacies of synapses with an increased number of receptor slots will grow, while the efficacies of other synapses will shrink, as we discuss in the following.

## Competition for receptors induces transient heterosynaptic plasticity

During LTP and LTD, the number of PSD-95 proteins in the synapse, which we assume to form the slots for AMPARs, is increased and decreased, respectively (**Colledge et al., 2003**; **Lisman and Raghavachari, 2006**; **Ehrlich et al., 2007**; **Meyer et al., 2014**). Importantly, these changes in slot numbers are mirrored by corresponding adjustments of synaptic AMPAR numbers leading to long

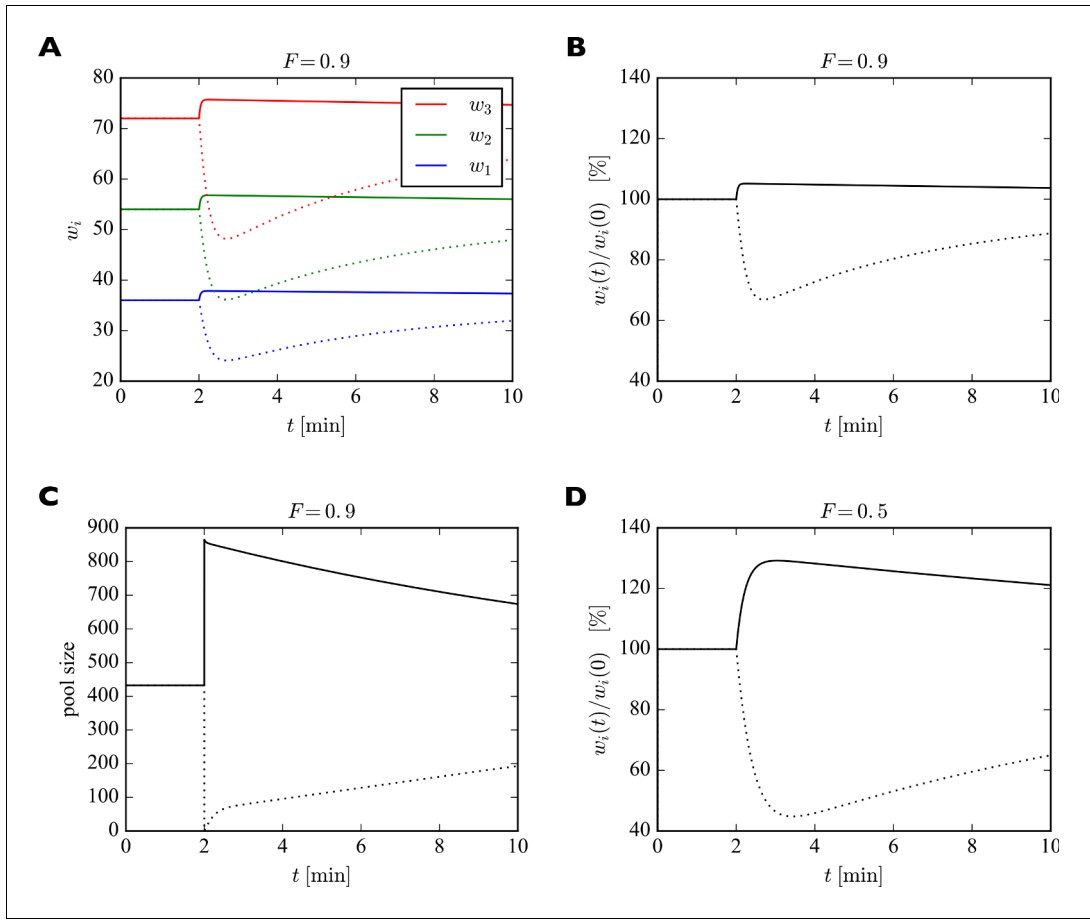

**Figure 4.** Effect of sudden change of the pool size $p$ on synaptic efficacies. (**A**) After 2 min, the pool size is either doubled (solid curves) or set to zero (dotted curves). In response, the synaptic efficacies are scaled multiplicatively as receptors are redistributed through the system. Doubling the receptor pool has a relatively weak effect in this example, as the system starts with a high filling fraction of 0.9, meaning that 90% of the slots are already filled at the beginning and there are few empty slots to which the additional receptors can bind. (**B**) Same as A. but showing relative change in synaptic efficacies, which is identical for all synapses. (**C**) Change in pool size. After the sudden increase or decrease in pool size at 2 min, there is first a rapid relaxation of the pool size followed by a much slower return towards the original value. (**D**) Same as B. but for a filling fraction of $1/2$. The smaller filling fraction leads to bigger relative changes of the synaptic efficacies. Parameters used were: $\beta = 1/43\,\text{s}^{-1}$, $\delta = 1/14\,\text{min}^{-1}$. The desired relative pool size was set to $\phi = 2.67$. The production rate $\gamma$ and attachment rate $\alpha$ were calculated according to **Equation 12** and **Equation 13**, respectively.

DOI: https://doi.org/10.7554/eLife.37836.007

lasting changes in synaptic efficacies. This suggests such modifications in AMPAR slot numbers as a central mechanism for memory storage. Therefore, we now investigate how the addition or removal of receptor slots in some synapses alters the efficacies of other synapses in the local group. We find that the model gives rise to a form of heterosynaptic plasticity, since all synapses are competing for a limited number of receptors inside the extrasynaptic receptor pool.

For illustration purposes we consider a piece of dendrite with four synaptic inputs (**Figure 5**). At the beginning of the simulation, the number of slots in the four synapses are 20, 40, 60, and 80. We start the system in its steady state with a filling fraction $F = 0.5$ and a relative pool size of $\phi = 2.67$. After 2 min we instantaneously increase the number of slots in the first (blue) and third (red) synapse by 100% (**Figure 5A**). Subsequently, the system settles into a new (transient) equilibrium (**Figure 5A**). While $w_1$ and $w_3$ increase, the number of receptors in synapses 2 and 4 slightly decrease, although their numbers of slots have not changed. This behavior corresponds to a form of heterosynaptic plasticity where synapses grow at the expense of other synapses and is due to the approximately constant

number of receptors on a fast time scale. Note that the sum of synaptic efficacies is not perfectly constant, however. The increase in synaptic efficacies $w_1$ and $w_3$ is bigger than the decrease of synaptic efficacies $w_2$ and $w_4$. The bigger the size of the pool, the stronger is this effect. Close to perfect balancing of synaptic weights would require $p \ll W$. *Figure 5B* shows the relative changes of efficacies of the synapses undergoing homosynaptic LTP (blue curve corresponding to $w_1$ and $w_3$ in **A**) vs. heterosynaptic LTD (green curve corresponding to $w_2$ and $w_4$ in **A**) and the pool (black curve).

What determines the magnitude of the heterosynaptic change? We can calculate this analytically by using the above short-term approximation $F^*$ for constant receptor number. Before plasticity induction, the synaptic efficacy of a synapse in equilibrium is given by $w_i = Fs_i = (s_i/S)W$. The induction of homosynaptic plasticity in other synapses changes the total number of available receptor slots and we denote the new number of slots $S'$. Shortly after homosynaptic plasticity induction the synaptic efficacies of a synapse that did not undergo homosynaptic plasticity will be approximately $w_i^* = F^* s_i = (s_i/S')W^*$ as receptors are redistributed through the system. Therefore, the relative

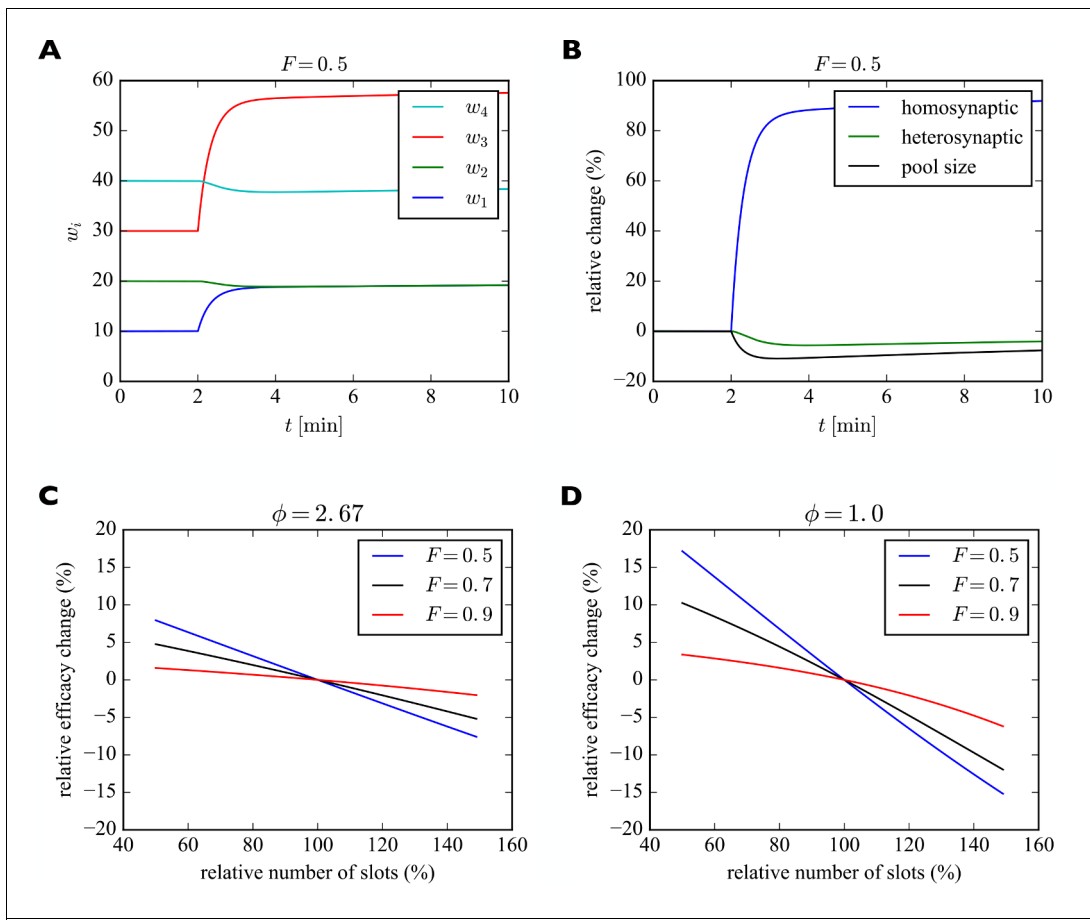

**Figure 5.** Induction of transient heterosynaptic plasticity. (**A**) Illustration of transient heterosynaptic plasticity. After 2 min, the number of slots in synapses 1 and 3 is increased instantaneously. The system quickly reaches a new (transient) equilibrium, where the non-stimulated synapses 2 and 4 are slightly weakened. At the same time, the number of receptors in the pool is reduced. Parameters were: $\beta = 1/43\,\mathrm{s}^{-1}$, $\delta = 1/14\,\mathrm{min}^{-1}$. The filling fraction was set to $F = 0.5$ and the relative pool size was set to $\phi = 2.67$. The production rate $\gamma$ and attachment rate $\alpha$ were calculated according to *Equation 12* and *Equation 13*, respectively. (**B**) Time course of relative changes in synaptic efficacies due to homosynaptic and heterosynaptic plasticity for the experiment from A. (**C**) Approximate maximum relative change of synaptic efficacy due to heterosynaptic plasticity as a function of the number of receptor slots after homosynaptic plasticity induction for different filling fractions. (**D**) Same as C but for a smaller relative pool size of $\phi = 1.0$. See text for details.

DOI: https://doi.org/10.7554/eLife.37836.008

heterosynaptic change of such a synapse is given by $(w_i^\star - w_i)/w_i = (F^\star - F)/F$ as long as the total number of receptors has not changed much.

In *Figure 5C* we plot this relative change in synaptic efficacy due to heterosynaptic plasticity as a function of the total number of receptor slots following homosynaptic plasticity induction for different filling fractions. The relative pool size is assumed to be $\phi = 2.67$. First, we can observe that reductions in the total number of slots due to homosynaptic LTD cause heterosynaptic LTP. Conversely, increases in the total number of slots due to homosynaptic LTP cause heterosynaptic LTD. Second, the amount of heterosynaptic plasticity depends on the filling fraction prior to plasticity induction. Specifically, a high filling fraction of 0.9 leads to weaker heterosynaptic plasticity.

*Figure 5D* shows the analogous solution for the case of a smaller receptor pool. Here we set the relative pool size to $\phi = 1.0$. Everything else is identical to *Figure 5C*. The scarcely filled pool strongly amplifies the heterosynaptic plasticity effect. When, e.g., new slots are added in this case, the synapses can recruit fewer receptors from the small receptor pool and the heterosynaptic effect on other synapses becomes bigger. A large receptor pool essentially functions as a buffer shielding synapses from heterosynaptic plasticity. Consistent with **C**, larger filling fractions again lead to less heterosynaptic plasticity.

Importantly, these effects are inherently transient. Over a sufficiently long time, the system will settle into a new (true) equilibrium, where every synapse has the same filling fraction $F$ determined by the rate constants $\alpha, \beta, \gamma, \delta$ as described above. The approach towards this new true equilibrium can be seen most easily in *Figure 5B*, where the relative change of synaptic efficacy due to heterosynaptic plasticity (green curve) slowly decays towards zero. The new equilibrium will be stable unless the numbers of slots in the synapses change again. In the particular example of *Figure 5A,B*, synapses 2 and 4 slowly return to their original efficacies, while synapses 1 and 3 remain permanently strengthened due to their increased number of slots. This effect might explain the often transient nature of heterosynaptic plasticity observed in experiments, e.g., *Abraham and Goddard (1983)*.

We therefore note the following additional predictions of the model (compare *Table 2*, *Heterosynaptic Plasticity I, II*): First, the amount of heterosynaptic plasticity is inversely related to the size of the local receptor pool and the filling fraction. Second, heterosynaptic plasticity is only transient.

Another mechanism for producing a heterosynaptic effect is changing the transition rates $\alpha$ and $\beta$ in a synapse-specific fashion. For example, increasing $\alpha$ for some synapses will attract additional receptors to these synapses and lead to a heterosynaptic removal of receptors from the remaining synapses and the receptor pool. A more complete model of homosynaptic LTP that includes a transient synapse-specific change in $\alpha$ and induces heterosynaptic LTD is discussed next.

## Time course of homosynaptic LTP and accompanying heterosynaptic LTD

The assumption of a sudden increase in slot numbers from the last section is helpful for mathematical analysis but does not reflect biological reality well. Receptor slots need to be trafficked and integrated into the PSD, which cannot happen instantaneously. In fact, modifications in PSD-95 protein number after plasticity induction are known to take several minutes (*Colledge et al., 2003*; *Ehrlich et al., 2007*; *Meyer et al., 2014*). In general, the induction of LTP is a complex process unfolding across multiple time scales. Here we propose and analyze a more realistic model of homosynaptic LTP and the accompanying heterosynaptic LTD. The model incorporates a synapse-specific transient increase in the insertion rate $\alpha$ of a potentiating synapse and a rapid and pronounced increase of its number of slots followed by a gradual decay back to a sustained elevated level. Thus, both the insertion rates $\alpha_i$ and the slot numbers $s_i$ are now considered a function of time. Formally, in order to do so we replace *Equations 1 and 2* by:

$$\dot{w}_i(t) = -\beta w_i(t) + \alpha_i(t)p(t)(s_i(t) - w_i(t)), \ i = 1, \ldots, N \tag{16}$$

$$\dot{p}(t) = -\delta p(t) + \gamma + \sum_i \beta w_i(t) - \sum_i \alpha_i(t)p(t)(s_i(t) - w_i(t)), \tag{17}$$

where we have introduced synapse specific insertion rates $\alpha_i$ and made the time dependence of the various quantities explicit.

We model the transient increase in $\alpha$ as a linear increase to four times the original value within 17 s followed by a linear decrease back to the original value over two minutes. This time course roughly corresponds to the one reported for calcium calmodulin kinase II (CaMKII) activation by Lee and colleagues (*Lee et al., 2009*), essential for LTP induction and maintenance (*Malenka et al., 1989*). CaMKII activation leads to the phosphorylation of many synaptic target proteins including the AMPAR auxiliary protein Stargazin which in turn increases the number of stabilized receptors in the synapse (*Opazo et al., 2010*). Thus, we assume here that increased CaMKII activation observed experimentally drives up the insertion rate $\alpha$. For the time course of the insertion of receptor slots, no direct measurements exist to our knowledge. Therefore, we make the simplifying assumption that the number of slots is related to the change in size of the dendritic spine, which was also measured by Lee and colleagues (*Lee et al., 2009*). We model their data as a sigmoidal increase to five times the original spine volume over the course of two minutes followed by an exponential decay to two times the original spine volume over the course of around twenty minutes (time constant of 5 min). We model the change in the number of receptor slots to scale with the $2/3$ power of the change in spine volume, assuming scaling with the surface area rather than the volume of the spine. The filling fraction was set to $F = 0.9$ and the relative pool size to $\phi = 2.67$. The results are shown in *Figure 6*.

*Figure 6A* shows the time course of the relative change of the insertion rate $\alpha$ and the number of receptor slots of the potentiated synapses. *Figure 6B* shows the time course of synaptic efficacies. At around 4 min the number of slots of the stimulated synapses peaks. Thereafter, the number of slots and the synaptic efficacies of the stimulated synapses decay to their new equilibrium values and the size of the receptor pool slowly recovers. In this example with a high filling fraction of $F = 0.9$ and a relative pool size of $\phi = 2.67$ the heterosynaptic effect is very small. This can also be seen in *Figure 6C*, which shows the relative changes in the synaptic efficacies and the pool size as a function of time. As in the previous section, the amount of heterosynaptic plasticity depends on the filling fraction and the relative size of the receptor pool, however. This is illustrated in *Figure 6D*, where we consider a smaller filling fraction of $F = 0.5$ and a smaller relative pool size of $\phi = 1.0$. This leads to a strong depletion of the receptor pool and a large heterosynaptic depression effect.

To quantify this effect, we systematically vary the relative pool size $\phi$ and filling fraction $F$ and observe the peak relative changes in synaptic efficacies during homosynaptic LTP and heterosynaptic LTD (*Figure 6E,F*). We find that a small pool size strongly reduces the peak homosynaptic LTP and greatly increases the peak heterosynaptic LTD. Furthermore, both homosynaptic LTP and heterosynaptic LTD tend to be reduced by a high filling fraction. These results are consistent with those from *Figure 5*.

In addition to these already noted effects on heterosynaptic plasticity, this implies another prediction of the model regarding homosynaptic plasticity (compare *Table 2*, *Homosynaptic*): The amount of short-term homosynaptic plasticity expression is modulated by the pool size and the filling fraction.

The changes in efficacies of synapses whose number of receptor slots are unaltered in *Figures 4*, *5* and *6* are only transient. In the following, we will study the long-term behavior of the model on the time scale associated with receptor externalization and internalization to determine how long it takes for the system to reach its (new) stable fixed point given by $w_i^\infty$ and $p^\infty$.

## Approach to the steady state is governed by externalization and internalization rates

To study the system's approach to its long-term steady state we again make use of the separation of time scales argument. Specifically, we assume that the fast dynamics of receptor exchanges between the pool and the synapses quickly reaches its equilibrium before the total number of receptors can change much due to receptor externalization and internalization. For this analysis we return to the original formulation of the model. Specifically, the change in the total receptor number from *Equation 1* and *Equation 2* is approximated by:

$$\dot{R} = \dot{p} + \dot{W} = -\delta p + \gamma \approx -\delta p^* + \gamma , \tag{18}$$

where we have replaced the current pool size $p$ with its steady state value $p^*(R) = R - W^*(R)$ for a constant number of receptors in the system. Using $F^*(R) \equiv W^*(R)/S$ we arrive at:

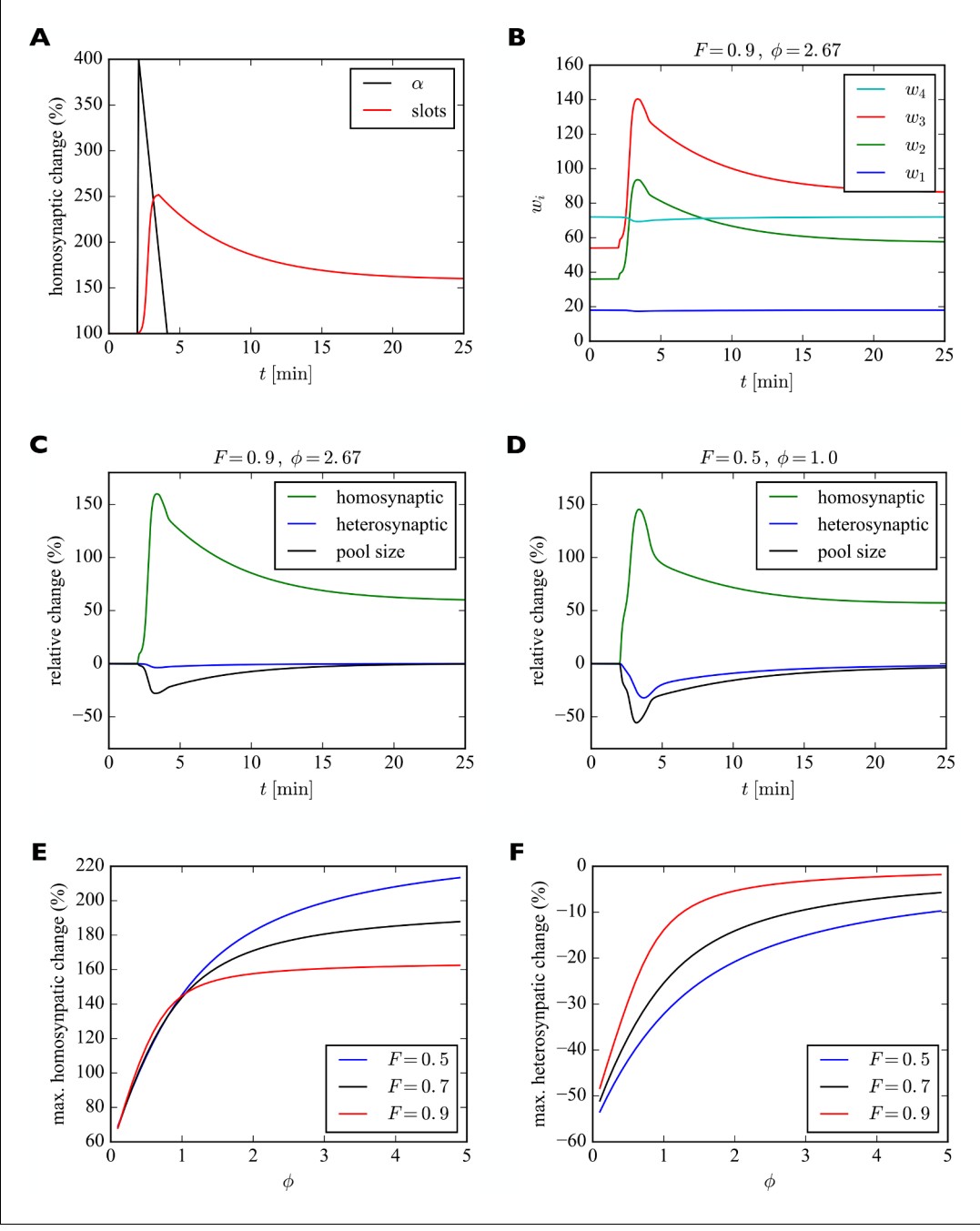

**Figure 6.** Model of homosynaptic LTP accompanied by heterosynaptic LTD. After 2 min, LTP is induced in synapses 2 and 3. This entails a transient synapse-specific change in the insertion rate $\alpha$ of these synapses and a gradual change in their slot numbers. (A) Time course of relative change of receptor insertion rate $\alpha$ and slot numbers of stimulated synapses undergoing homosynaptic LTP. (B) Time course of synaptic efficacies for a filling fraction of $F = 0.9$ and a relative pool size of $\phi = 2.67$. Only a very small amount of heterosynaptic LTD can be observed in unstimulated synapses 1 and 4. (C) Relative change of synaptic efficacies and pool size due to homosynaptic LTP and heterosynaptic LTD in B as a function of time. (D) Same as C but for a smaller filling fraction of $F = 0.5$ and a smaller relative pool size of $\phi = 1.0$. Note the smaller transient increase in efficacy of potentiated synapses (compare peaks of green curves in C and D) and the increased amount of heterosynaptic LTD (compare troughs of blue curves). (E, F) Maximum amount of homosynaptic LTP (E) and heterosynaptic LTD (F) as a function of relative pool size $\phi$ for three different filling fractions.

DOI: https://doi.org/10.7554/eLife.37836.009

$$\dot{R} = \gamma + \delta F^*(R)S - \delta R. \tag{19}$$

For small numbers of receptors in the system, that is, $R$ close to zero, the steady state filling fraction $F^*(R)$ will be close to zero so that $\dot{R} \approx \gamma$. In contrast, for high numbers of receptors and the filling fraction close to its long-term steady-state value $F$ we find:

$$\frac{1}{\delta}\dot{R} \approx \frac{\gamma}{\delta} + FS - R, \tag{20}$$

indicating that $R$ will exponentially approach its steady state value of $\gamma/\delta + FS$ with the time constant $1/\delta$. This behavior is illustrated in *Figure 7*. The simulated piece of dendrite has a total of 10 000 receptor slots and is initialized with different receptor numbers. We plot the numerical solution of *Equation 19* for different initial numbers of receptors in the system. For low receptor numbers, the growth rate of $R$ is approximately $\gamma$ (compare dotted line). For a filling fraction close to its final steady-state value, $R$ exponentially converges to its steady state of $\gamma/\delta + FS$ with time constant $\delta^{-1}$.

## Smaller spontaneous synaptic efficacy fluctuations in larger synapses

Our analysis of the differential equation model above is suitable for studying the average behavior of the system for large numbers of receptors. However, small synapses may only have a few receptors inside them and the effects of stochastic fluctuations may become substantial. To quantify the size of such fluctuations of bound receptor numbers we have developed a stochastic version of the model that explicitly simulates the stochastic binding and unbinding, internalization and externalization of individual receptors (see Materials and methods). We use the model to study the fluctuations of synaptic efficacies under basal steady state conditions. This allows us to quantify the size of synaptic efficacy fluctuations due to the fast exchange of AMPARs between synapses and the receptor pool.

For illustration, we consider a local group of 7 synapses with 1, 2, 5, 10, 20, 50, and 100 slots, respectively. We quantify the size of fluctuations of synaptic efficacies using the coefficient of

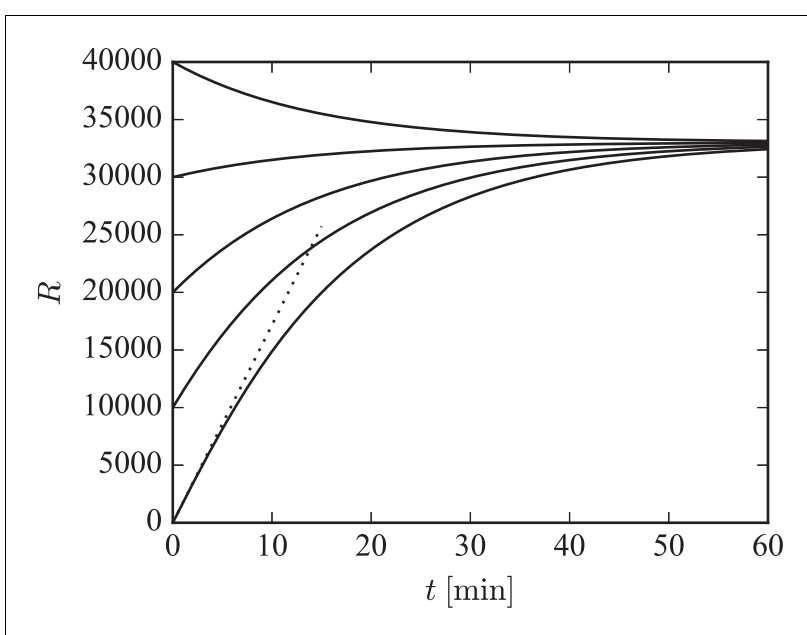

**Figure 7.** Illustration of long-term behavior under the separation of time scales assumption. Parameters were: $\beta = 1/43\,\mathrm{s}^{-1}$, $\delta = 1/14\,\mathrm{min}^{-1}$. The desired relative pool size was set to $\phi = 2.67$ and the desired filling fraction to $F = 0.9$. The production rate $\gamma$ and attachment rate $\alpha$ were calculated according to *Equation 12* and *Equation 13*, respectively. The steady-state total number of receptors in this example is given by $R^\infty = (1 + \phi)FS = 33\,030$.
DOI: https://doi.org/10.7554/eLife.37836.010

variation (CV), which is defined as the standard deviation of the fluctuating number of receptors bound inside a synapse divided by the time average of the number of receptors bound inside this synapse. A high CV indicates strong relative fluctuations of the synapse's efficacy.

*Figure 8A* shows the numbers of receptors bound in each synapse as a function of time in one example simulation of 10 min. Parameters are as given in *Table 1* with the filling fraction set to $F = 0.5$ and the relative pool size set to $\phi = 2.67$. *Figure 8B* shows an example for a much higher filling fraction of $F = 0.9$. Fluctuations are greatly attenuated.

*Figure 8C* plots the logarithm of the CV of the number of receptors as a function of the logarithm of the average number of receptors per synapse, which is given by the product of the theoretical filling fraction $F$ and the number of receptor slots $s_i$ of the synapse $i$. Data are shown for three different filling fractions obtained by increasing $\alpha$, the rate of receptors binding to receptor slots, while setting $\gamma$ to maintain a constant relative pool size of $\phi = 2.67$. The linear relationships evident in the log-log plot indicate a power law scaling. We fit power law functions of the form $\mathrm{CV} = a(Fs)^b$ to the

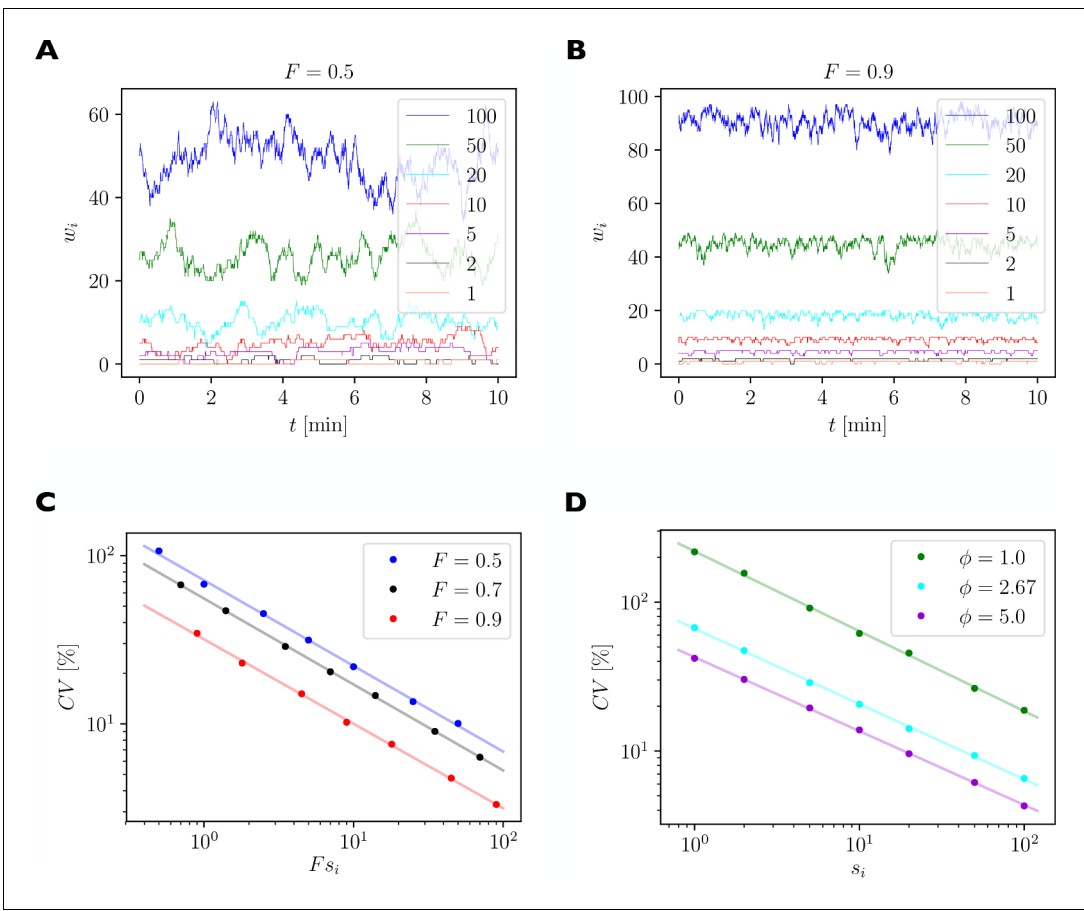

**Figure 8.** Quantification of spontaneous synaptic efficacy fluctuations due to the fast exchange of receptors between synapses and the receptor pool. (A) Example simulation of a piece of dendrite with 7 synapses during 10 min of simulated time. The number of receptor slots in each synapse is given in the legend. The relative pool size was set to $\phi = 2.67$ and the filling fraction was set to $F = 0.5$ by choosing the binding rate to receptor slots $\alpha$ via *Equation 13*. (B) Same as A. but for a higher filling fraction of $F = 0.9$. (C) Size of synaptic efficacy fluctuations as measured by the coefficient of variation (CV) as a function of the steady state number of receptors in each synapse, which is given by the product of the filling fraction $F$ and the number of slots $s_i$ in synapse $i$. The relative pool size was set to $\phi = 2.67$. Data points represent averages over 10 simulations of 30 min simulated time each. Lines represent linear fits through the data points in double log space. (D) CV as a function of steady state number of receptors for different relative pool sizes $\phi$ achieved by holding the binding rate to receptor slots $\alpha$ fixed and varying the externalization rate $\gamma$.
DOI: https://doi.org/10.7554/eLife.37836.011

data (solid lines). Parameters of the fits are given in *Table 3* and indicate slopes of around $-1/2$, i.e. the CV declines approximately with one over the square root of the average number of bound receptors. Specifically, small synapses exhibit substantial fluctuations of their efficacies with CVs of up to 100%, while fluctuations are greatly attenuated in strong synapses. For different filling fractions, the curves are shifted vertically such that fluctuations are particularly strong for a filling fraction of 0.5 and are reduced for higher filling fractions.

*Figure 8D* considers the case where the rates $\alpha, \beta, \delta$ are held constant and the externalization rate $\gamma$ is varied to achieve different relative pool sizes $\phi \in \{1.0, 2.67, 5.0\}$. Specifically, to achieve a particular relative pool size $\phi$ we set:

$$\gamma = \delta \left( S\phi - \frac{\beta}{\alpha} \right). \tag{21}$$

The change in $\gamma$ also leads to different filling fractions in the three cases, see *Equation 7*. The results in *Figure 8D* show that an increased pool size will dampen spontaneous fluctuations of synaptic efficacies, while a reduced pool size promotes stronger fluctuations. We again fit power law functions to the data. Parameters of the fits are given in *Table 4*. Taken together, these results imply another prediction of the model (compare *Table 2*, *Fluctuations*): Small synapses undergo relatively larger spontaneous efficacy fluctuations, which are attenuated by a large pool size.

In conclusion, the spontaneous exchange of synaptic building blocks between synapses and dendritic pool leads to substantial fluctuations in synaptic efficacies. This finding is reminiscent of the surprisingly large spontaneous fluctuations in spine sizes in the absence of activity-dependent synaptic plasticity observed recently (*Dvorkin and Ziv, 2016*; *Shomar et al., 2017*; *Ziv and Brenner, 2018*).

## Discussion

The detailed molecular mechanisms underlying different forms of synaptic plasticity are complex. Recent years have seen enormous progress in identifying many of the relevant molecules and signaling pathways. This rapid development is in stark contrast to the simplistic and often purely phenomenological descriptions of synaptic plasticity used in most neural network models. While highly simplified mathematical models have been essential for relating synaptic plasticity 'rules' to learning processes at the network level, a full understanding of synaptic plasticity requires the development of more elaborate models that do justice to the complexities of synaptic plasticity at the molecular scale (*Bhalla, 2011*; *Bhalla, 2014*; *Tsodyks et al., 1998*; *Urakubo et al., 2008*). Here we have taken a step in this direction.

Hebbian learning tends to lead to runaway growth of synaptic efficacies if not counteracted by competitive or homeostatic mechanisms. To be effective, these compensatory mechanisms must act fast enough so they can catch up with changes induced by Hebbian learning (*Zenke et al., 2013*; *Chistiakova et al., 2015*). Prominent candidate mechanisms are synaptic normalization and heterosynaptic plasticity (*Lynch et al., 1977*). The idea has a long history. Synapses on the dendritic tree compete for a limited supply of synaptic building blocks such that when some synapses grow, they have to do so at the expense of other synapses (*von der Malsburg, 1973*; *Lynch et al., 1977*; *Antunes and Simoes-de-Souza, 2018*). However, until recently, the lack of knowledge on the nature

**Table 3.** Fitting results from the stochastic version of the model, cf. *Figure 8C*.

The externalization rate $\gamma$ and the attachment rate of receptors to slots $\alpha$ are set to obtain different filling fractions while maintaining a relative pool size of $\phi = 2.67$. Parameters $\beta$ and $\delta$ are as in Tab. 1. $a$ and $b$ give the parameters of the power law fits.

| F | $\alpha [\mathrm{min}^{-1}]$ | $\gamma [\mathrm{min}^{-1}]$ | **Scale factor** $a$ | **Exponent** $b$ |
|---|---|---|---|---|
| 0.5 | 0.0056 | 12.1 | 71.4 | $-0.52$ |
| 0.7 | 0.0093 | 9.4 | 55.6 | $-0.51$ |
| 0.9 | 0.0278 | 6.7 | 31.8 | $-0.50$ |

DOI: https://doi.org/10.7554/eLife.37836.012

**Table 4.** Fitting results from the stochastic version of the model, cf. *Figure 8D*.
The attachment rate of receptors to slots is chosen as $\alpha = 0.0093\,\mathrm{min}^{-1}$ to obtain a filling fraction of 0.7 for a relative pool size of $\phi = 2.67$. Parameters $\beta$ and $\delta$ are as in Tab. 1. $\gamma$ is varied to obtain different relative pool sizes $\phi$ and filling fractions $F$.

| Relative pool size $\phi$ | $\gamma\,[\mathrm{min}^{-1}]$ | $F$ | Scale factor $a$ | Exponent $b$ |
|---|---|---|---|---|
| 1.0 | 2.67 | 0.20 | 92.6 | −0.54 |
| 2.67 | 25.1 | 0.7 | 55.4 | −0.51 |
| 5.0 | 56.4 | 0.84 | 39.1 | −0.50 |

DOI: https://doi.org/10.7554/eLife.37836.013

and the timescales of the molecular processes taking place at synapses did not allow for realistic modeling of such a competition for synaptic resources. Here we have presented a concrete model with a fast normalization of the efficacies of a neuron's afferent synapses based on this competition for synaptic resources such as AMPA-type glutamate receptors.

Our model makes several contributions. First, it formalizes the idea of a fast synaptic normalization based on a competition for dendritic resources in an abstract and analytically tractable model. Second, analysis of the model reveals that under the given assumptions, normalization should act multiplicatively, such that relative strengths of synapses are maintained. Multiplicative normalization rules have been used in neural network models for a long time but usually in an *ad hoc* fashion. Our model supports the idea that a fast multiplicative normalization may in fact be biologically plausible. Third, the model naturally gives rise to a transient form of homeostatic heterosynaptic plasticity where synapses grow in efficacy at the expense of other synapses. Fourth, the model quantifies how the amount of heterosynaptic plasticity depends on the size of the local receptor pool and the filling fraction of receptor slots. It thereby reveals a fundamental trade-off: the smaller the pool of available receptors, the more pronounced the heterosynaptic plasticity. In other words, neurons can limit heterosynaptic plasticity effects, but this comes at the price of having to maintain a big receptor pool. Similarly, the model predicts that a larger receptor pool attenuates spontaneous fluctuations in synaptic efficacies, which are particularly strong for small synapses. In the following we discuss how this prediction and others summarized in *Table 2* could be tested.

## How to test the model's predictions

The first prediction of the model is that synapses in a local group have identical filling fractions, see *Equation 7*. That is, under basal conditions the same percentage of receptor slots should be filled in these synapses on average. Testing this prediction requires measuring both the number of receptor slots and the number of filled receptor slots for a local group of individual synapses. This could be achieved using a quantitative super resolution approach such as dual-color direct stochastic optical reconstruction microscopy (dSTORM). For a given dendrite, one would have to quantify the number of AMPARs and PSD-95 proteins in a local group of synapses under basal conditions. Our prediction is that the ratio of AMPARs to PSD-95 proteins should be similar in all the synapses. As a corollary, we predict that the relative efficacies of two synapses from a local group should be identical to their relative slot numbers. Testing this hypothesis requires measuring the slot numbers and synaptic efficacies of two synapses from a local group. Specifically, the efficacies of a group of synapses could initially be measured using local glutamate uncaging. Subsequently the number of PSD-95 could be assessed using dSTORM. This second approach seems rather challenging, however, as one would have to find in the fixed sample the exact dendrite and specific spines that were stimulated during live-imaging. A second corollary of the model's first prediction is that if any of the transition rates changes, e.g., the rate at which receptors unbind from receptor slots, the filling fractions and synaptic efficacies are scaled by the same factor. Testing this prediction can be achieved by interventions that alter the transition rates. Activation of CaMKII leads to the phosphorylation of the AMPAR auxiliary subunit Stargazin increasing its affinity to PSD-95 (*Hafner et al., 2015*). Thus one could induce a global activation of CaMKII in the neurons (chemical-LTP), fix the cells immediately after, and perform dual-color dSTORM for PSD-95 proteins and AMPARs. When comparing basal state to

chemical-LTP, the ratio of PSD-95 proteins to AMPARs should decrease by the same factor for all synapses.

The model's second prediction can be tested in a similar way. We predict that manipulating the size of the local receptor pool leads to a multiplicative rescaling of the efficacies of the local group of synapses. To test this prediction, the size of the local receptor pool has to be altered, e.g., by triggering externalization of additional receptors. This could be achieved by treating neurons with TNF-$\alpha$ for instance (*Zhao et al., 2010*). Subsequently the efficacies of the local group of synapses have to be monitored. These efficacies should scale by the same factor.

Another prediction of the model is that the amount of heterosynaptic plasticity is inversely related to the size of the local receptor pool. The most direct way of testing this prediction is to manipulate the local receptor pool as suggested above while inducing homosynaptic plasticity in a subset of synapses and measuring the amount of heterosynaptic plasticity in other synapses of the local group. In fact, this set of experiments could resemble the ones performed by Oh and colleagues but adding a TNF-$\alpha$ condition (*Oh et al., 2015*).

The transient nature of heterosynaptic plasticity predicted by the model can be tested more easily. It merely requires the induction of homosynaptic plasticity in a subset of synapses in the local group while monitoring the time course of heterosynaptic plasticity in the remaining synapses. Specifically, the time course of recovery from heterosynaptic plasticity should be close to the internalization rate $\delta$ of AMPARs.

The model's prediction of an influence of the size of the receptor pool on the expression of homosynaptic plasticity requires manipulating the size of the local receptor pool and subsequently inducing homosynaptic plasticity. For example, the peak change in synaptic efficacy during LTP induction should be bigger when the receptor pool has been increased than when it has been depleted prior to LTP induction.

Finally, the model predicts that synapses exhibit spontaneous fluctuations in synaptic efficacies due to the dynamic exchange of receptors with the local receptor pool. It predicts that these fluctuations, as measured by the coefficient of variation (CV), scale approximately as one over the square root of the synapses' average efficacies. Testing this prediction can be achieved by repeated measurements of the synaptic efficacy of single synapses in the absence of any plasticity induction using glutamate uncaging.

## Dendritic morphology and local production

We have assumed that the basal transition rates for receptors attaching and detaching to and from slots are identical for all synapses and that the receptors are distributed homogeneously inside the pool. These assumptions are essential for the multiplicative behavior of the model. If, in contrast, the distribution of receptors across the dendritic tree were very inhomogeneous, this would, all else being equal, correspond to different pool sizes in different parts of the dendritic tree, leading to different filling fractions across the dendritic tree.

Properly distributing synaptic building blocks across the dendritic tree is a formidable task (*Williams et al., 2016*). Specifically, if receptors were only produced at a single site corresponding to the neuron cell body (or soma) and spreading from this point source according to slow transport processes then one would expect a high concentration of receptors close to the soma and a low concentration far away from it. This would, all else being equal, lead to large receptor pools close to the soma and small receptor pools far away from it. *Earnshaw and Bressloff (2008)* have presented such a model. They consider a long dendrite and diffusion of receptors from the soma along this dendrite leading to a high concentration of receptors close to the soma and a small concentration far away from it. In contrast, our model considers a local piece of dendrite, where the concentration of receptors can be assumed to be approximately constant. Therefore, our model does not attempt to make predictions regarding scaling of synaptic efficacies at the **global** level of a neuron's entire dendritic tree. Earnshaw and Bressloff conclude from their model that 'it does not appear possible to obtain a global multiplicative scaling' of synaptic efficacies just by changing reaction rates. This conclusion rests on the fact that the distribution of receptors along their simulated dendrite is inhomogeneous. Specifically, Earnshaw and Bressloff assume that protein synthesis occurs mostly at the soma, which leads to an approximately exponential decay of the concentration of receptors towards the distal end of the dendrite. This assumption failed to be confirmed experimentally and has in fact been contradicted by Tao–Cheng and colleagues who found a homogeneous distribution of

AMPARs at the neuron surface along the dendritic arbor of hippocampal cultured neurons (*Tao-Cheng et al., 2011*). Earnshaw and Bressloff also cite a study by *Adesnik et al. (2005)* to support the idea of an inhomogeneous distribution of AMPARs. They used ANQX (a modified version of DNQX) known at that time as an AMPAR antagonist (*Chambers et al., 2004*) to monitor synaptic AMPAR exchange after specific inactivation of the surface population. They measured a significantly slower recovery of AMPAR current in dendrites compared to the somatic region. Thus, they concluded that AMPARs are mainly exocytosed at the somatic extracellular membrane and trafficked distally through lateral diffusion. However, since then DNQX has also been shown to act on kainate and NMDA receptors. Additionally, DNQX effects on AMPARs appear to depend on the composition of AMPAR complexes and in particular the type of auxiliary subunits associated with those receptors (*Maclean and Bowie, 2011*; *Greger et al., 2017*). Since the concentration of receptors between somatic and dendritic membranes appears to be fairly homogeneous, it might be that the actual composition of the receptors varies between those two compartments. In this case, global multiplicative scaling is to be expected in the model of Earnshaw and Bressloff as well. Hence, we believe that our model using minimal assumptions and being restricted to a single dendritic segment with multiple dendritic spines is in good accordance with the recent literature on AMPAR trafficking.

A uniform distribution of synaptic building blocks across the entire dendritic tree could be facilitated by local production of these building blocks across the dendritic tree. Local protein synthesis may therefore be essential for global multiplicative scaling behavior observed in biological experiments (*Turrigiano et al., 1998*). More specifically, synthesis of AMPAR subunits happens inside the cell at the endoplasmic reticulum membrane (ER). This synthesis of proteins seems to occur in a burst fashion in local 'hot spots' distributed across the dendritic tree (*Katz et al., 2016*). Importantly, however, newly synthesized receptors are not necessarily immediately trafficked to the cell surface and in fact a large fraction are distributed across and maintained inside the ER compartment constituting an intracellular pool of receptors waiting to be exocytosed (*Greger et al., 2002*). Thus, the distribution of receptors in the ER may already be more homogeneous than hot spot synthesization would suggest. Furthermore, once released from the ER into the cytoplasm, fast distribution of receptors along microtubules could lead to a rather homogeneous distribution inside the cytoplasm, from where the receptors would be trafficked to the surface. Thus, bursty translation at hotspots inside the ER may still allow for a homogeneous distribution of receptors at the cell surface. We therefore predict that local production of synaptic building blocks across the dendritic tree contributes to their uniform distribution, which in turn might allow global multiplicative scaling behavior and the maintenance of relative strengths of synapses. This could be tested, for example, by specifically blocking local production of synaptic building blocks, which should make their distribution across the dendritic tree less homogeneous and lead to systematic inhomogeneities in synaptic efficacies across the dendritic tree.

In this context it is also interesting to note that at least one form of heterosynaptic plasticity tends to be induced locally (*De Roo et al., 2008*; *Losonczy et al., 2008*; *Li et al., 2016b*), that is, at neighboring synapses. Such local action is readily expected in our model if competition for synaptic building blocks is restricted to a *local* pool such as a section of a dendritic branch, with comparatively slow trafficking of building blocks between adjacent pools.

### Detailed descriptions of AMPAR trafficking and diffusion

The stochastic version of our model describes individual binding and unbinding events of AMPARs to receptor slots, but it does not describe in detail the paths taken by individual AMPARs during their diffusion in the cell membrane. This is a gross simplification, but it facilitates mathematical analysis. More elaborate models were conceived to describe the movement of individual receptors inside the dendritic branch and the PSD (*Earnshaw and Bressloff, 2006*; *Czöndör et al., 2012*; *Li et al., 2016a*). Such models can incorporate, e.g., the detailed spine geometry or effects of protein crowding.

### Control of transition rates

Apart from our experiments on modeling LTP, where we introduced a transient and synapse-specific increase of the rate at which receptors bind to slots, we have kept all transition rates constant throughout this paper. In reality, we expect the various transition rates to be flexibly controlled to

allow for robust and efficient functioning of the neuron, allowing it to cope with various perturbations. Indeed, constructing a model to describe these various regulatory processes will be an important challenge for the future. Furthermore, AMPARs can be in different states expected to have different transition rates. Specifically, AMPAR complexes containing various sets of auxiliary subunits are very likely to co-exist at the neuron surface (*Schwenk et al., 2012*). Since only a couple of auxiliary subunits have binding domains for PSD-95, multiple types of AMPARs with different $\alpha$ and $\beta$ parameters could be considered. These topics are left for future work.

## Slot production and removal

In future work, it will also be interesting to consider changes to slot numbers in more detail. We simulated increases in slot numbers of individual synapses in the context of LTP. Obviously, however, the building blocks of these 'slots' also have to be produced, transported, and inserted into synapses, which could be based on similar mechanisms as we have postulated for receptors. Furthermore, slots are also degraded and have to be replaced. In fact, the alternative interpretation of our model discussed in the beginning of the Results section already describes how PSD-95 slots are produced (or degraded) and bind to (or detach from) slots for these receptor slots ('slots-for-a-slot' interpretation). Future work should aim for a model that more fully describes the interactions of AMPARs (and other types of receptors), various TARPs such as stargazin, MAGUK proteins such as PSD-95, and neuroligins as well as their production and trafficking. Along these lines, it will also be interesting to consider the mechanisms underlying different stages of LTP and LTD in more detail.

## Modeling slow homeostatic synaptic scaling

The model could also be extended to capture slow homeostatic synaptic scaling processes (*Turrigiano et al., 1998*; *Ibata et al., 2008*). In the simplest case, a sensor for the average neural activity of the neuron would drive the production of receptors and/or slots in a homeostatic fashion, such that if, e.g., the average neural activity falls below a target level or range, then receptor and/or slot production are increased to drive up excitatory synaptic efficacies. Such a model would naturally explain the multiplicative behavior of homeostatic synaptic scaling (*Turrigiano et al., 1998*). Obviously, the activity sensor could also sense the average activity in a local neighborhood through a diffusive mechanism (*Sweeney et al., 2015*). Furthermore, instead of homeostatically regulating firing rates, the amount of afferent drive to the neuron or to the local population could be controlled (*Savin et al., 2009*), or even other measures of neural and synaptic activity could be used. Finally, all these ideas are not mutually exclusive. It seems likely that neurons control both their firing rate distributions and their amounts of excitatory and inhibitory afferent drive through a combination of different intrinsic and synaptic plasticity mechanisms on different time scales.

## Receptor subunit composition

Finally, not all AMPARs are created equal. Depending on the composition of subunits, AMPARs have distinct properties in terms of, e.g., calcium permeability and trafficking (see *Henley and Wilkinson, 2016*) for a recent review). A more complete model should incorporate the diversity of AMPARs (or even other receptor types) and their properties.

## Conclusion

In conclusion, our model offers a parsimonious explanation for a transient form of homeostatic heterosynaptic plasticity and fast local synaptic normalization, which it predicts to be multiplicative. It therefore supports the use of such rules in neural network models. The model also reveals a fundamental trade-off between the size of the local receptor pool and the amount of heterosynaptic plasticity. This trade-off is akin to a common logistics problem: how much to produce and store of a particular resource in order to (a) minimize production costs and storage space while (b) limiting the risk of running out of this resource? Arguably, efficient neural functioning requires solving a plethora of related logistics problems with respect to production, transport, and storage of various 'goods' and supply of the necessary energy for all these processes. We feel that the time is ripe for a concerted effort to study individual neurons and the entire nervous system from such a *neurologistics* perspective.

## Materials and methods

### Simulation software

The simulation software was written in Python and is available at: https://github.com/triesch/synaptic-competition (*Triesch and Vo, 2018*); copy archived at https://github.com/elifesciences-publications/synaptic-competition).

Differential equations were discretized with the Euler method.

The stochastic version of the model was simulated using the Gillespie algorithm (*Gillespie, 1976*). Stochastic reactions were defined for receptors entering or leaving each of the seven synapses and for being added or removed from the receptor pool. This gave rise to a total of 16 possible 'reactions' occurring with different probabilities per unit time depending on the current state of the system, that is, how many receptors are bound in each synapse and reside in the pool. Stochastic simulations were validated against the differential equation model to verify that their average behavior matched that of the differential equation model in different situations.

### Calculation of the short-term filling fraction

We exploit the separation of time scales between fast receptor binding and unbinding from slots and slow externalization and internalization of receptors. On the fast time scale, the processes of internalization and externalization can be ignored. Removing the corresponding terms in *Equation 2*, we again look for the steady state solution by setting the time derivatives of $w_i$ and $p$ to zero and summing over $i$. This leads to the following quadratic equation for $W^*$, the steady state number of bound receptors in the short-term approximation (which must not be confused with the long-term steady state solution $W^\infty$ of the full system):

$$W^{*2} - W^*\left(S + R + \frac{\beta}{\alpha}\right) + RS = 0. \tag{22}$$

We introduce $\rho \equiv \beta/\alpha$ as the ratio of the rates through which receptors leave and enter the synaptic slots. Using this, the two solutions of *Equation 22* are given by:

$$W^*_{1,2} = \frac{1}{2}(S + R + \rho) \pm \sqrt{\frac{1}{4}(S + R + \rho)^2 - RS}. \tag{23}$$

The '+' solution is not biologically meaningful, since it leads to $W^* \geq S$ or $W^* \geq R$ (see Appendix), so that the desired steady state solution of the short-term approximation is given by:

$$W^* \equiv W^*_2 = \frac{1}{2}(S + R + \rho) - \sqrt{\frac{1}{4}(S + R + \rho)^2 - RS} \tag{24}$$

and the corresponding short-term steady-state filling fraction is $F^* = W^*/S$. In the full system, this solution is assumed only transiently, because receptors can still enter and leave the system. If the number of receptors were held constant, then $F^*$ and $W^*$ would describe the stable solution on long time scales.

### Sensitive reaction of the short-term filling fraction to changes in reaction rates when number of receptors matches number of slots

We are interested in how the short-term filling fraction $F^*$ changes, when the reaction rates $\alpha$ and $\beta$ or their ratio $\rho \equiv \beta/\alpha$ change. Formally, we consider the partial derivative of the short-term filling fraction $F^* = W^*/S$ with respect to $\rho$. Using *Equation 24* we find:

$$\frac{\partial F^*}{\partial \rho} = \frac{1}{S}\frac{\partial W^*}{\partial \rho} = \frac{1}{S}\left[\frac{1}{2} - \frac{R + S + \rho}{4\sqrt{\frac{1}{4}(R + S + \rho)^2 - RS}}\right]. \tag{25}$$

As can be seen in *Figure 3C,D*, the most extreme slope is obtained at $\rho = 0$. There the derivative simplifies to:

$$\left.\frac{\partial F^*}{\partial \rho}\right|_{\rho=0} = \frac{1}{2S}\left(1 - \frac{R+S}{R-S}\right). \tag{26}$$

For $R = S$ the slope diverges, that is, the short term filling fraction reacts extremely sensitively to small changes in $\rho$ when $\rho$ is close to zero.

## Additional information

### Funding

| Funder | Grant reference number | Author |
|---|---|---|
| Johanna Quandt Foundation | | Jochen Triesch |
| European Molecular Biology Organization | ALTF 1095-2015 | Anne-Sophie Hafner |
| Alexander von Humboldt-Stiftung | 3.3-1184902-FRA-HFST-P | Anne-Sophie Hafner |

The funders had no role in study design, data collection and interpretation, or the decision to submit the work for publication.

### Author contributions

Jochen Triesch, Conceptualization, Software, Formal analysis, Validation, Investigation, Visualization, Methodology, Writing—original draft, Project administration, Writing—review and editing; Anh Duong Vo, Software, Formal analysis, Validation, Investigation, Visualization, Methodology; Anne-Sophie Hafner, Conceptualization, Investigation, Writing—original draft, Writing—review and editing

### Author ORCIDs

Jochen Triesch (iD) http://orcid.org/0000-0001-8166-2441
Anne-Sophie Hafner (iD) https://orcid.org/0000-0003-4416-7307

### Decision letter and Author response

Decision letter https://doi.org/10.7554/eLife.37836.017
Author response https://doi.org/10.7554/eLife.37836.018

## Additional files

### Supplementary files

• Transparent reporting form
DOI: https://doi.org/10.7554/eLife.37836.014

Program code of the model is publicly available at: https://github.com/triesch/synaptic-competition (copy archived at https://github.com/elifesciences-publications/synaptic-competition).

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

## Appendix 1

DOI: https://doi.org/10.7554/eLife.37836.015

### The '+' solution from *Equation 23* is not biologically meaningful.

We show that the '+' solution from *Equation 23* is not biologically meaningful. To see this, first note that $W_1 \leq W_2$. Furthermore, any meaningful solution $W$ must fulfill $W \leq R$ and $W \leq S$, that is, the number of receptors bound to slots cannot be bigger than the total number of receptors or the total number of slots. If the smaller solution $W_1$ does not meet both criteria, then the larger $W_2$ cannot meet them either. So we assume in the following that $W_1$ meets both these criteria so that $W_1 \leq \min\{R, S\}$. Our argument uses Vieta's formulas for the quadratic *Equation 22*:

$$W_1 + W_2 = R + S + \rho \quad \text{and} \quad W_1 W_2 = RS.$$

Using the second formula we can write:

$$RS = W_1 W_2 \leq \min\{R, S\} W_2,$$

from which follows that:

$$W_2 \geq \frac{RS}{\min\{R, S\}}.$$

In the case that $R > S$, this leads to $W_2 \geq R$. The only biologically meaningful solution to this is the equality $W_2 = R$. This is the extreme case where all receptors are bound in slots and no receptors remain in the pool. With Vieta's second formula we see that in this case $W_1 = S$. Plugging both results into Vieta's first formula, we see that this solution requires $\rho = 0$, which in turn requires $\beta = 0$. In this case, no receptors would ever leave synapses.

The case $R < S$ leads to $W_2 \geq S$. The only biologically meaningful solution to this is the equality $W_2 = S$. This is the extreme case where all slots are filled with receptors. Using Vieta's formulas again leads to the uninteresting requirement $\rho = 0$ for this solution.

Finally, the case $R = S$ leads to $R = S = W_1 = W_2$ and also requires $\rho = 0$. In summary, the '+' solution in *Equation 23* only admits the extreme solutions $W = S$ or $W = R$ requiring $\rho = 0$ (and therefore $\beta = 0$), which are not biologically meaningful.

