## [Decision Letter]

[Editors’ note: the authors were asked to provide a plan for revisions before the editors issued a final decision. What follows is the editors’ letter requesting such plan.]

Thank you for sending your article entitled "Competition for synaptic building blocks shapes synaptic plasticity" for peer review at *eLife*. Your article has been reviewed by three peer reviewers, one of whom is a member of our Board of Reviewing Editors, and the evaluation has been overseen by Eve Marder as the Senior Editor.

The reviewers have discussed the reviews with one another and the Reviewing Editor has drafted this decision to help you prepare a revised submission.

In particular, the relevance, novelty and main conclusions of the manuscript may be undermined by previous work that was not cited:

Earnshaw and Bressloff, 2006 and Earnshaw and Bressloff,.2008.

The second paper concludes, "even given a number of simplifying assumptions, it does not appear possible to obtain a global multiplicative scaling of synaptic receptor numbers along a dendrite from a simple up or down regulation of constitutive recycling." This contradicts the present study's main conclusion. See reviewer #2's comments for additional details.

Please examine these previous studies carefully and prepare a response, outlining whether these disparities can be addressed and whether the present study offers insights, results or an approach that is more relevant than this previous work.

Reviewer #1:

This paper proposes a novel mathematical model that explains the heterosynaptic plasticity in local dendrite after the induction of LTP and/or LTD. The model comprises four processes at the cellular and molecular level, which are the binding and unbinding of receptors to and from slots on postsynaptic membrane, and the addition and removal of free receptors to and from the local pool in the dendrite. The model is biophysically sound and analysed in detail to reveal how it predicts/accounts for a variety of higher level effects. For example, by quantifying the number of filled slots and the size of local pool on short timescales (i.e. shorter that biosynthesis), the model successfully predicts that:

1) On short time scales the redistribution of receptors between synapses is multiplicative, a phenomenon commonly known as synaptic scaling.

2) The amount of heterosynaptic plasticity is inversely related to the size of local receptor pool.

Transient heterosynaptic plasticity has rarely been modelled in systematically from the perspective of receptor trafficking, or in a way that highlights the role of timescale separation. This paper is accessible to experimentalists and will aid in forming experimental predictions. In particular, the model suggests that synaptic scaling can happen locally, and independently of biosynthesis on the timescales it is typically studied in experiments. In other words, synaptic scaling is simply an unavoidable consequence of having a pool of receptors with constitutive trafficking.

Issues:

To improve the presentation, the authors should reduce the number of predictions they made (some of them are almost tautologies!) to make the most important prediction stand out. These and other specific issues that should be clarified/addressed are outlined below:

- Subsection “Formulation of the model”: Need to briefly explain the physical meaning of involving *p* in the term *αp(s_i_ – w_i_)*.

- Table 2 Scaling II: The prediction is only true in steady-state (or on average).

- Subsection “Competition for Synaptic Building Blocks Induces Multiplicative Scaling”, fifth paragraph: Three predictions may be reduced to one otherwise it could be difficult to grasp your key point. They all come from the same Equation 7 in fact, so you could keep one most important prediction in Table 2 and leave others as part of your analysis in this section.

- Subsection “Fast redistribution of receptors between synapses is multiplicative”, first paragraph: should read "the steady state solution of *fast time scale*".

- Subsection “Fast redistribution of receptors between synapses is multiplicative”, second paragraph: "As above" is inaccurate. *F** here is clearly different from *F* in Equation 7, as γ=0 can no longer serve as a denominator.

- Subsection “Fast redistribution of receptors between synapses is multiplicative”, second paragraph: Articulate more explicitly. This is one of the most important predictions and therefore deserves a detailed explanation. For example, add the mathematical relationship between the size of local receptor *p* and the synaptic efficacy *w* in the quasi steady-state:

w*=p*p+p*si

- Subsection “Competition for receptors induces transient heterosynaptic plasticity”, first paragraph: Motivate it better for changing the number of slots. It is a bit out of the blue here, because until now s has never been assumed to be a temporal variable in your model.

- Subsection “Time course of homosynaptic LTP and accompanying heterosynaptic LTD”, first paragraph: Explain/justify "not reflect biological reality well".

- Suggest a clear distinction between α modulation and *s* modulation.

Reviewer #2:

This study fleshes out the implications of an existing idea, that multiplicative synaptic scaling of synapses could be due to competition for shared synaptic resources, identified by the authors as e.g. AMPARs.

I very much like the approach but am wary of two of the main conclusions and predictions (and also the novelty of the findings), for the following reasons:

1) A similar model was studied by Earnshaw and Bressloff across two papers (not cited in the current study):

Earnshaw and Bressloff, 2006. and Earnshaw and, 2008. The 2008 paper is particularly relevant because they studied a model that included receptor diffusion along the dendrite, where synapses competed for the same shared pool of receptors. They concluded that "even given a number of simplifying assumptions, it does not appear possible to obtain a global multiplicative scaling of synaptic receptor numbers along a dendrite from a simple up or down regulation of constitutive recycling.". The current study may have not found this for two reasons: first the Bressloff model is nonlinear, and second the Bressloff model includes space. Can the authors comment on this discrepancy? Does it undermine their whole study?

2) I find it surprising that the model predicts (Equation 6) that the total number of receptors in the pool is independent of synapse number, slot number, and receptor binding and unbinding rates (α and β). The maths makes sense – due to linearity of the model – but it would be nice if the authors could comment on the likely validity of this prediction/assumption if non-linearities were to be introduced. For example, there may be two types of receptor state within the synapse, trapped and not trapped.

Reviewer #3:

This paper represents an important intermediate level of modeling between purely abstract views of normalization and homeostasis of synapse strength and detailed biological modeling, something that is still impossible at this scale but which is being attempted in the context of the individual spine.

The authors do a good job of reaching up (top-ward) to the abstract formulation but make less of an effort to reach down (bottom-ward) to the biology. The down-reach is more difficult and will necessarily be speculative but I would encourage the authors to make this effort. This represents the essence of such a level-of-investigation bridging model: from the problem level (normalization), to algorithms, to implementations.

"In the limit of large receptor numbers" – please justify. What are the estimated numbers? What numbers are required to avoid substantial variability due to stochastics? There is brief discussion of this in the Discussion section that could be moved up.

Predictions are the eventual goal of modeling and should be given more attention. Please move the predictions of Table 2 into a major textual section and indicate how each prediction could currently (or with some imagined technical advance in the future) be tested experimentally.

Implementations determines algorithms determines problems (inverse of the Marr procedure). In this case, what is the role of catabolism of damaged receptors? Are most receptors actually returned to the pool or do many need to be replaced? What are estimates of the metabolic requirements for such replacement? How are receptors within the dendritic pool mobilized into spines? What is the involvement of endoplasmic reticulum, rough ER?

"Spreading from [nucleus by] slow diffusion process" is not really accurate; what is putative interplay of microtubules and actin vs. role of diffusion in bringing receptors into play? How might these factors relate to the role of pool mobilization in synaptic tagging and capture (STC)? Does the sudden increase in pool size in Figure 4 represent a 'capture' event?

[Editors’ note: formal revisions were requested, following approval of the authors’ plan of action.]

Thank you for sending a revision plan for your article entitled "Competition for synaptic building blocks shapes synaptic plasticity" for peer review at *eLife*. We are pleased to accept a final, revised version of your manuscript conditional on the essential changes outlined in the revision plan and the comments below.

Your revision plan for this article has been evaluated by 3 peer reviewers, and the evaluation is being overseen by a Reviewing Editor and Eve Marder as the Senior Editor. In addition to the revisions you have outlined, we request, in line with reviewer comments, that you address the following important issues, which were raised in the evaluation:

- "The authors should make more explicit throughout the manuscript that the analysis does not predict global synaptic scaling. This is especially important in the discussion, which links their findings to the global synaptic scaling as studied by Turrigiano et al. It seems another component of the model would need to be added to make this link; perhaps something linking the receptor pools between dendrites, or between dendrites and the soma."

- "One small point - Bressloff et al didn't assume a non-uniform distribution of synaptic receptors, that was a prediction from the model and indeed was one of the reasons that they claimed multiplicative scaling was tricky."

- "Finally there are empirical reasons to challenge the authors' assumption that synaptic receptor expression is flat within even a single dendrite. This might be OK as a simplifying assumption for a local analysis but it needs to be put in context of experimental evidence e.g. Spruston and Burrone labs have found that synaptic protein content seems to decrease from proximal to distal portions of single dendrites in hippocampal pyramidal neurons (Menon et al, Neuron, 2013; Bloss et al, Neuron, 2016; Walker et al, PNAS, 2017)."

Please resubmit a revised manuscript with the changes outlined in your revision plan as well as addressing the comments above.

---

## [Author Response]

[Editors’ notes: the authors’ response after being formally invited to submit a revised submission follows.]

In particular, the relevance, novelty and main conclusions of the manuscript may be undermined by previous work that was not cited:Earnshaw and Bressloff, 2006 and Earnshaw and Bressloff, 2008.The second paper concludes; "even given a number of simplifying assumptions, it does not appear possible to obtain a global multiplicative scaling of synaptic receptor numbers along a dendrite from a simple up or down regulation of constitutive recycling." This contradicts the present study's main conclusion. See reviewer #2's comments for additional details.Please examine these previous studies carefully and prepare a response, outlining whether these disparities can be addressed and whether the present study offers insights, results or an approach that is more relevant than this previous work.

We were unaware of the papers by Earnshaw and Bressloff. Thank you for pointing them out to us and our apologies for this oversight. The 2006 paper is concerned with a detailed model of receptor trafficking inside a single dendritic spine. The 2008 paper simplifies the 2006 model (e.g. no distinction between different receptor types) but at the same time extends it to a long one-dimensional dendrite and describes the surface diffusion of receptors along this dendrite. Our model is in fact intermediate in that we consider a *local* piece of dendrite, where the concentration of receptors in the local pool can be treated as approximately constant. Therefore, our model does not speak to the question of *global* multiplicative scaling, but it does predict a *local* multiplicative scaling. Importantly, research since 2008 suggests that some of the assumptions of Earnshaw and Bressloff (2008) need to be revisited, which could invalidate some of their conclusions. We elaborate on this in our detailed response to reviewer 2.

"The authors should make more explicit throughout the manuscript that the analysis does not predict global synaptic scaling. This is especially important in the Discussion, which links their findings to the global synaptic scaling as studied Okby Turrigiano et al. It seems another component of the model would need to be added to make this link; perhaps something linking the receptor pools between dendrites, or between dendrites and the soma."

We have made this clear throughout the paper now and specifically discuss local vs. global synaptic scaling in the Discussion (section “Dendritic morphology and local production”).

"One small point – Bressloff et al. didn't assume a non-uniform distribution of synaptic receptors, that was a prediction from the model and indeed was one of the reasons that they claimed multiplicative scaling was tricky."

Yes, we are aware of this, but admit that our wording did not make this clear. This has been corrected.

"Finally there are empirical reasons to challenge the authors' assumption that synaptic receptor expression is flat within even a single dendrite. This might be OK as a simplifying assumption for a local analysis but it needs to be put in context of experimental evidence e.g. Spruston and Burrone labs have found that synaptic protein content seems to decrease from proximal to distal portions of single dendrites in hippocampal pyramidal neurons (Menon et al., Neuron, 2013; Bloss et al., Neuron, 2016; Walker et al., PNAS, 2017)."

We thank the reviewer for pointing out this aspect of neuronal function. Indeed, we are aware that synapses along the full dendritic arbor of a neuron will differ in their composition. In fact, many labs have shown that ion channels in particular can be specifically expressed in definite regions of the dendritic arbor of CA1 pyramidal neurons. Local protein synthesis that we propose as a compensatory mechanism for the point source supply of synaptic proteins is unlikely to concern all synaptic proteins. Also, some proteins may exclusively be synthesized in specific dendritic regions. Thus, some elements of the synapse will show non-homogeneous distribution along the dendritic arbor. This is the main reason, as the reviewer points out, why we consider a small portion of dendrite and not the full dendritic arbor in the current manuscript.

In the present study, what concerns us the most is the homogeneous distribution of surface AMPARs in a small portion of a dendrite. Toa–Cheng and colleagues have described such a distribution for GluA2-containing AMPARs at the neuron surface along the dendritic arbor of cultured hippocampal neurons (Toa-Cheng et al., 2011).

In Menon et al., 2013 the authors describe in Figure 3 that the number of AMPARs per perforated synapses increases, as dendrites are further away from the soma. Interestingly, this effect is not observed in non-perforated synapses that are by far the most abundant type of synapse on CA1 pyramidal neurons (ratio perforated to nonperforated synapses range from 0.1 to 0.25 (Nicholson et al., 2006)). This suggests that perforated synapses might represent an intermediate plasticity state associated with a transient increase in AMPAR insertion rate α.

Reviewer #1:This paper proposes a novel mathematical model that explains the heterosynaptic plasticity in local dendrite after the induction of LTP and/or LTD. The model comprises four processes at the cellular and molecular level, which are the binding and unbinding of receptors to and from slots on postsynaptic membrane, and the addition and removal of free receptors to and from the local pool in the dendrite. The model is biophysically sound and analysed in detail to reveal how it predicts/accounts for a variety of higher level effects. For example, by quantifying the number of filled slots and the size of local pool on short timescales (i.e. shorter that biosynthesis), the model successfully predicts that:1) On short time scales the redistribution of receptors between synapses is multiplicative, a phenomenon commonly known as synaptic scaling.2) The amount of heterosynaptic plasticity is inversely related to the size of local receptor pool.Transient heterosynaptic plasticity has rarely been modelled in systematically from the perspective of receptor trafficking, or in a way that highlights the role of timescale separation. This paper is accessible to experimentalists and will aid in forming experimental predictions. In particular, the model suggests that synaptic scaling can happen locally, and independently of biosynthesis on the timescales it is typically studied in experiments. In other words, synaptic scaling is simply an unavoidable consequence of having a pool of receptors with constitutive trafficking.

We agree and have combined some predictions.

Issues:To improve the presentation, the authors should reduce the number of predictions they made (some of them are almost tautologies!) to make the most important prediction stand out. These and other specific issues that should be clarified/addressed are outlined below:- Subsection “Formulation of the model”: Need to briefly explain the physical meaning of involving p in the term αp(s_i_ – w_i_).

Explanation has been added.

- Table 2 Scaling II: The prediction is only true in steady-state (or on average).

Correct, has been clarified.

- Subsection “Competition for Synaptic Building Blocks Induces Multiplicative Scaling”, fifth paragraph: Three predictions may be reduced to one; otherwise it could be difficult to grasp your key point. They all come from the same Equation 7 in fact, so you could keep one most important prediction in Table 2 and leave others as part of your analysis in this section.

Agreed. We have combined the scaling predictions into one.

- Subsection “Fast redistribution of receptors between synapses is multiplicative”, first paragraph: should read "the steady state solution of fast time scale".

Agreed. Change has been made.

- Subsection “Fast redistribution of receptors between synapses is multiplicative”, second paragraph: "As above" is inaccurate. F* here is clearly different from F in Equation 7, as γ=0 can no longer serve as a denominator.

Has been rephrased. The “As above” was meant to refer to the property of all synapses having the identical filling fraction.

- Subsection “Fast redistribution of receptors between synapses is multiplicative”, second paragraph: Articulate more explicitly. This is one of the most important predictions therefore deserves a detailed explanation. For example, add the mathematical relationship between the size of local receptor p and the synaptic efficacy w in the quasi steady-state:

w*=p*p+p*si

We have elaborated on this.

- Subsection “Competition for receptors induces transient heterosynaptic plasticity”, first paragraph: Motivate it better for changing the number of slots. It is a bit out of the blue here, because until now s has never been assumed to be a temporal variable in your model.

We have added some sentences and made clear references to literature pointing to the tight structure-function relationship between synaptic metrics and synaptic strength and plasticity (1, 2). In this context we believe modulating slot numbers (as a rough approximation of PSD size) will be justified as an initial attempt to simulate forms of homosynaptic plasticity.

- Subsection “Time course of homosynaptic LTP and accompanying heterosynaptic LTD”, first paragraph: Explain/justify "not reflect biological reality well".

Agreed, this deserves a more careful explanation and in particular references to the time scale of modification of PSD-95 protein number in synapses during plasticity (1) as well as justifications of how CaMKII activation could impact α (insertion rate) while *s* (number of slots) may initially remain unchanged.

- Suggest a clear distinction between α modulation and s modulation.

We are not entirely sure what you mean here. Obviously, a time dependence could be introduced only for α or only for s. We feel that discussing (and simulating) all these cases separately is not likely to add important insights. However, we have rephrased to suggest that these two kinds of modulation can be considered independently of each other.

Reviewer #2:This study fleshes out the implications of an existing idea, that multiplicative synaptic scaling of synapses could be due to competition for shared synaptic resources, identified by the authors as e.g. AMPARs.I very much like the approach but am wary of two of the main conclusions and predictions (and also the novelty of the findings), for the following reasons:1) A similar model was studied by Earnshaw and Bressloff across two papers (not cited in the current study):Earnshaw and Bressloff, 2006. and Earnshaw and, 2008. The 2008 paper is particularly relevant because they studied a model that included receptor diffusion along the dendrite, where synapses competed for the same shared pool of receptors. They concluded that "even given a number of simplifying assumptions, it does not appear possible to obtain a global multiplicative scaling of synaptic receptor numbers along a dendrite from a simple up or down regulation of constitutive recycling.". The current study may have not found this for two reasons: first the Bressloff model is nonlinear, and second the Bressloff model includes space. Can the authors comment on this discrepancy? Does it undermine their whole study?

Thank you for pointing out these studies, which we had simply overlooked. The most important difference is that Earnshaw and Bressloff consider a long dendrite and the diffusion along this long dendrite, while our model considers a local piece of dendrite, where the concentration of receptors can be assumed to be approximately constant. Therefore, our model does not attempt to make predictions regarding scaling at the *global* level of synapses across a neuron's entire dendritic tree. Furthermore, Earnshaw and Bressloff's conclusion that “it does not appear possible to obtain a global multiplicative scaling” rests on the assumption that the distribution of receptors along their simulated dendrite is inhomogeneous. Specifically, Earnshaw and Bressloff assume that protein synthesis occurs mostly at the soma and that the concentration of receptors falls off approximately exponentially towards the distal end of their dendrite.

To the best of our knowledge, this assumption failed to be confirmed experimentally and has in fact been contradicted by a study from the lab of Thomas Reese. Toa–Cheng and colleagues found homogenous distribution of GluA2-containing AMPARs at the neuron surface along the dendritic arbor of hippocampal cultured neurons (3). Earnshaw and Bressloff made the inhomogeneous distribution of AMPARs assumption based a study by Adesnik and colleagues (4). In this study, the authors used ANQX (a modify version of DNQX) known at that time as an AMPAR antagonist, to monitor synaptic AMPAR exchange after specific inactivation of the surface population (5). In their experiments, they measured a significantly slower recovery of AMPAR current in dendrites compare to the somatic region. Thus they concluded that AMPARs are mainly exocytosed at the somatic extracellular membrane and trafficked distally through lateral diffusion. However, DNQX has been shown since then not to be specific to AMPAR but to also act on kainate and NMDA receptors. Additionally DNQX and CNQX inhibition of AMPARs seems to be dependent on the composition of AMPAR complexes and in particular the type of auxiliary subunits associated with those receptors (6, 7). While the concentration of receptors between somatic and dendritic membranes appears to be fairly homogenous, it might be that the actual composition of the receptors varies between those two compartments. This could explain the observation by Adesnik and colleagues in the absence of an inhomogeneous receptor distribution. And in this case global multiplicative scaling is to be expected in the model of Earnshaw and Bressloff as well. Thus, we believe that our model using minimal assumptions and being restricted to a single dendritic segment with multiple dendritic spines is in better accordance with the recent literature on AMPAR trafficking. We now discuss all these matters in the manuscript.

2) I find it surprising that the model predicts (Equation 6) that the total number of receptors in the pool is independent of synapse number, slot number, and receptor binding and unbinding rates (α and β). The maths makes sense – due to linearity of the model – but it would be nice if the authors could comment on the likely validity of this prediction/assumption if non-linearities were to be introduced. For example, there may be two types of receptor state within the synapse, trapped and not trapped.

Strictly speaking, the model is non-linear, because there are product terms of the state variables (*p x w_i_*) in Equations 1 and 2. Interestingly, a slightly modified model (still non-linear), where receptors in slots can be internalized directly with the same rate δ without first going through the pool does not have the surprising property you mention. This model seemed less biologically plausible to us. Other, more complex, models with different receptor states are an interesting topic for future work (and Earnshaw and Bressloff have already looked at this to some extent). However, without having implemented and analyzed these models in detail it's hard to foresee how their predictions might differ. Similarly, AMPAR complexes containing various sets of auxiliary subunits are also very likely to co-exist at the neuron surface (8). Since only a couple of auxiliary subunits have binding domains for PSD-95, it might also be that multiple types of slots with different α and β parameters could be considered. Nevertheless, those topics are better left for future work. We have added a brief discussion of this.

Reviewer #3:This paper represents an important intermediate level of modeling between purely abstract views of normalization and homeostasis of synapse strength and detailed biological modeling, something that is still impossible at this scale but which is being attempted in the context of the individual spine.The authors do a good job of reaching up (top-ward) to the abstract formulation but make less of an effort to reach down (bottom-ward) to the biology. The down-reach is more difficult and will necessarily be speculative but I would encourage the authors to make this effort. This represents the essence of such a level-of-investigation bridging model: from the problem level (normalization), to algorithms, to implementations."In the limit of large receptor numbers" – please justify. What are the estimated numbers? What numbers are required to avoid substantial variability due to stochastics? There is brief discussion of this in the Discussion section that could be moved up.

We have implemented a stochastic version of the model that simulates stochastic transitions of individual receptors using the Gillespie algorithm. This has allowed us to quantify the size of synaptic efficacy fluctuations and precisely answer all these questions. We have added this material to the manuscript (one additional figure, one additional co-author). A new insight (not really unexpected, though) coming from these simulations is that small synapses can have substantial fluctuations in their efficacies due to fast receptor trafficking even under basal conditions. In contrast, large synapses have comparatively stable efficacies. Another insight is that the size of these fluctuations depends on the filling fraction of the synapses.

Predictions are the eventual goal of modeling and should be given more attention. Please move the predictions of Table 2 into a major textual section and indicate how each prediction could currently (or with some imagined technical advance in the future) be tested experimentally.

Having a table summarizing the major predictions was actually our attempt to make them more prominent. We have kept the table but also added a text section to the Discussion elaborating how the various predictions could be tested.

Implementations determines algorithms determines problems (inverse of the Marr procedure). In this case, what is the role of catabolism of damaged receptors? Are most receptors actually returned to the pool or do many need to be replaced? What are estimates of the metabolic requirements for such replacement? How are receptors within the dendritic pool mobilized into spines? What is the involvement of endoplasmic reticulum, rough ER?

The current version of the model does not consider intracellular trafficking per se. Thus involvement of the endoplasmic reticulum is beyond the scope of this study. Regarding receptor damage it is known that AMPAR half-life is rather short (~2 days – while the average half-life of neuronal proteins is 5-7 days) (9, 10). So AMPARs are constantly degraded and replaced by newly synthetized proteins. But, in any case, damaged receptors would have to be endocytosed into intracellular compartments to undergo lysosome-mediated degradation. We now briefly discuss the role of the ER in protein synthesis in the context of the hotspot translation events (see above).

The dendritic pool of AMPARs is composed of receptors that have been produced or recycled in intracellular compartments and externalized to the cell surface. After externalization AMPARs are diffusing at the neuron surface. AMPARs are mobilized into synapses mainly via binding to PSD-95 (11–13). At postsynaptic sites PSD-95 molecules are highly packed (~300 molecules per postsynaptic synaptic density) (14) and largely immobile (15). When a receptor enters a synapse binding to one or more immobile PSD-95 results in receptor immobilization. We have added corresponding text to the manuscript.

"Spreading from [nucleus by] slow diffusion process" is not really accurate; what is putative interplay of microtubules and actin vs. role of diffusion in bringing receptors into play? How might these factors relate to the role of pool mobilization in synaptic tagging and capture (STC)? Does the sudden increase in pool size in Figure 4 represent a 'capture' event?

Even the transport along microtubules may be viewed as a kind of diffusion process, where one single random walk “step” in the large-scale “diffusion” process is the transport along a long stretch of microtubule. Differing probabilities of forward vs. backward steps would lead to a biased random walk but a random walk nevertheless and therefore a form of diffusion. To avoid any confusion, we have rephrased to “slow transport process”. The sudden increase in pool size in Figure 4 was not intended to model a capture event as suggested in STC. However, the increase in slot numbers and binding rate to receptor slots used in Figure 6 could be interpreted as part of an STC event.

References:

1) Meyer D, Bonhoeffer T, Scheuss V. Article Balance and Stability of Synaptic Structures during Synaptic Plasticity. Neuron [Internet]. Elsevier; 2014;82(2):430–43. Available from: http://dx.doi.org/10.1016/j.neuron.2014.02.031

2) Bartol TM, Bromer C, Kinney J, Chirillo MA, Bourne JN, Harris KM, et al. Nanoconnectomic upper bound on the variability of synaptic plasticity. Elife [Internet]. 2015 [cited 2016 Jul 10];4:e10778.

3) Toa-Cheng J-H, Crocker VT, Winters CA, Azzam R, Chludzinski J, Reese TS. Trafficking of AMPA Receptors at Plasma Membranes of Hippocampal Neurons. J Neurosci. 2011;31(13):4834–43.

4) Adesnik H, Nicoll RA, England PM. Photoinactivation of Native AMPA Receptors Reveals Their Real-Time Trafficking. 2005;48:977–85.

5) Chambers JJ, Gouda H, Young DM, Kuntz ID, England PM. Photochemically Knocking Out Glutamate Receptors in Vivo. 2004;13886–7.

6) Maclean DM, Bowie D. Transmembrane AMPA receptor regulatory protein regulation of competitive antagonism : a problem of interpretation. 2011;22:5383–90.

7) Greger IH, Watson JF, Cull-candy SG. Review Structural and Functional Architecture of AMPA-Type Glutamate Receptors and Their Auxiliary Proteins. Neuron [Internet]. Elsevier Inc.; 2017;94(4):713–30.

8) Schwenk J, Harmel N, Brechet A, Zolles G, Berkefeld H, Müller CS, et al. High-resolution proteomics unravel architecture and molecular diversity of native AMPA receptor complexes. Neuron [Internet]. 2012 May 24

9) Cohen LD, Zuchman R, Sorokina O, Müller A, Dieterich DC, Armstrong JD, et al. Metabolic Turnover of Synaptic Proteins: Kinetics, Interdependencies and Implications for Synaptic Maintenance. PLoS One. 2013;8(5).

10) Dörrbaum AR, Kochen L, Langer JD, Schuman EM. Local and global influences on protein turnover in neurons and glia. Elife [Internet]. 2018;7:1–24.

11) Schnell E, Sizemore M, Karimzadegan S, Chen L, Bredt DS, Nicoll RA. Direct interactions between PSD-

95 and stargazin control synaptic AMPA receptor number. Proc Natl Acad Sci U S A. 2002 Oct 15

12) Bats C, Groc L, Choquet D. The interaction between Stargazin and PSD-95 regulates AMPA receptor surface trafficking. Neuron [Internet]. 2007 Mar 1 [cited 2014 Oct 26];53(5):719–34. Available from: http://www.ncbi.nlm.nih.gov/pubmed/17329211

13) Sumioka A, Yan D, Tomita S. TARP phosphorylation regulates synaptic AMPA receptors through lipid bilayers. Neuron [Internet]. 2010 Jun 10 [cited 2015 Feb 23];66(5):755–67.

14) Kim E, Sheng M. PDZ domain proteins of synapses. Nat Rev Neurosci [Internet]. 2004 Oct 1

15) Sturgill JF, Steiner P, Czervionke BL, Sabatini BL. Distinct domains within PSD-95 mediate synaptic incorporation, stabilization, and activity-dependent trafficking. J Neurosci. 2009;29(41):12845–54.